# Multilevel neural simulation-based inference

**Yuga Hikida**
Aalto University
yuga.hikida@aalto.fi

**Ayush Bharti**
Aalto University
ayush.bharti@aalto.fi

**Niall Jeffrey**
University College London
n.jeffrey@ucl.ac.uk

**François-Xavier Briol**
University College London
f.briol@ucl.ac.uk

## Abstract

Neural simulation-based inference (SBI) is a popular set of methods for Bayesian inference when models are only available in the form of a simulator. These methods are widely used in the sciences and engineering, where writing down a likelihood can be significantly more challenging than constructing a simulator. However, the performance of neural SBI can suffer when simulators are computationally expensive, thereby limiting the number of simulations that can be performed. In this paper, we propose a novel approach to neural SBI which leverages multilevel Monte Carlo techniques for settings where several simulators of varying cost and fidelity are available. We demonstrate through both theoretical analysis and extensive experiments that our method can significantly enhance the accuracy of SBI methods given a fixed computational budget.

## 1 Introduction

Simulation-based inference (SBI) [Cranmer et al., 2020] is a set of methods used to estimate parameters of complex models for which the likelihood is intractable but simulating data is possible. It is particularly useful in fields such as cosmology [Jeffrey et al., 2021], epidemiology [Kypraios et al., 2017], ecology [Beaumont, 2010], synthetic biology [Lintusaari et al., 2017], and telecommunications engineering [Bharti et al., 2022a], where models describe intricate physical or biological processes such as galaxy formation, spread of diseases, the interaction of cells, or propagation of radio signals.

For a long time, SBI was dominated by methods such as approximate Bayesian computation (ABC) [Beaumont et al., 2002, Beaumont, 2019], which compared summary statistics of simulations and of the observed data. However, SBI methods using neural networks to approximate likelihoods [Papamakarios et al., 2019, Lueckmann et al., 2019, Boelts et al., 2022], likelihood ratios [Thomas et al., 2022, Durkan et al., 2020, Hermans et al., 2020], or posterior distributions [Papamakarios and Murray, 2016, Lueckmann et al., 2017, Greenberg et al., 2019, Radev et al., 2022] are now quickly becoming the preferred approach. These *neural SBI* methods are often favoured because they allow for *amortisation* [Zammit-Mangion et al., 2024], meaning that they require a large offline cost to train the neural network, but once the network is trained, the method can rapidly infer parameters for new observations or different priors without requiring additional costly simulations. This is particularly useful when the simulator is computationally expensive, as it reduces the need to repeatedly run simulations for each new inference task, making the overall process significantly less costly.

Nevertheless, the initial training phase of neural SBI methods typically still requires a large number of simulations, preventing their application on computationally expensive (and often more realistic) models which can take hours of compute time for simulating a single data-point. Examples include most tsunami [Behrens and Dias, 2015], wind farm [Kirby et al., 2023], nuclear fusion [Hoppe et al., 2021] and cosmology [Jeffrey et al., 2025] simulators.

39th Conference on Neural Information Processing Systems (NeurIPS 2025).

One avenue to mitigate this issue is multi-fidelity methods [Peherstorfer et al., 2018]: we often have access to a sequence of simulators with increasing computational cost and accuracy which we may be able to use to refine existing methods based on a single simulator.

This setting is quite common. One example is simulators requiring the numerical solution of ordinary, partial, or stochastic differential equations, where the choice of mesh size or stepsize affects both the accuracy of the solution and the computational cost. Another example arises when modelling complex physical, chemical, or biological processes, where low-fidelity simulators can be obtained by neglecting certain aspects of the system. This is exemplified in Figure 1, where a high-fidelity cosmological simulation includes baryonic astrophysics and thus appears smoother than the low-fidelity simulation.

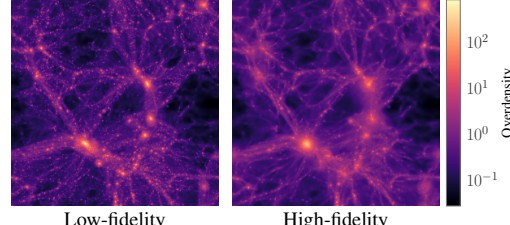

Figure 1: Low- and high-fidelity cosmological simulations from the CAMELS data [Villaescusa-Navarro et al., 2023] studied in Section 5.4.

The key idea behind multi-fidelity methods is to leverage cheaper, less accurate simulations to supplement the more expensive, high-fidelity simulations, ultimately improving efficiency without sacrificing accuracy. This has been popular in the emulation literature since the seminal work of Kennedy and O'Hagan [2000], but applications to SBI are much more recent and include Jasra et al. [2019], Warne et al. [2018], Prescott and Baker [2020, 2021], Warne et al. [2022], Prescott et al. [2024], who proposed multi-fidelity versions of ABC. More recently, [Krouglova et al., 2025] also proposed a multi-fidelity method to enhance neural SBI approximations of the posterior based on transfer learning. However, their method is not supported by theoretical guarantees.

In this paper, we propose a novel multi-fidelity method which is broadly applicable to neural SBI methods. Taking neural likelihood and neural posterior estimation as the main case studies, we show that our approach is able to significantly reduce the computational cost of the initial training through the use of multilevel Monte Carlo [Giles, 2015, Jasra et al., 2020] estimates of the training objective. The approach has strong theoretical guarantees; our main result (Theorem 1) directly links the reduction in computational cost to the accuracy of the low-fidelity simulators, and we demonstrate (in Theorem 2) how to best balance the number of simulations at each fidelity level in the process. Our extensive experiments on models from finance, synthetic biology, and cosmology also demonstrate the significant computational advantages provided by our method.

## 2 Background

We first recall the basics of SBI methods and the related works on reducing computational cost in Section 2.1, then provide a brief introduction to multilevel Monte Carlo in Section 2.2.

### 2.1 Simulation-based inference

Let $\{\mathbb{P}_\theta\}_{\theta \in \Theta}$ be a parametric family of distributions on some space $\mathcal{X} \subseteq \mathbb{R}^{d_\mathcal{X}}$ parameterised by $\theta \in \Theta \subseteq \mathbb{R}^{d_\Theta}$. We assume that this model is available in the form of a computer code, i.e. as a *simulator*, where simulating from $\mathbb{P}_\theta$ is straightforward, but the likelihood function $p(\cdot \mid \theta)$ associated with $\mathbb{P}_\theta$ is intractable. Simulators can be characterised by a pair $(\mathbb{U}, G_\theta)$, where $\mathbb{U}$ is a distribution (typically simple, such as a uniform or a Gaussian) on a space $\mathcal{U} \subseteq \mathbb{R}^{d_\mathcal{U}}$ which captures all of the randomness, and $G_\theta : \mathcal{U} \mapsto \mathcal{X}$ is a (deterministic) parametric map called the *generator*. Simulating $x \sim \mathbb{P}_\theta$ can be achieved by first simulating $u \sim \mathbb{U}$, and then applying the generator $x = G_\theta(u)$. In this paper, we consider Bayesian inference for the parameters $\theta$ of this simulator-based model given independent and identically distributed (iid) data $x^o_{1:m} = \{x^o_j\}^m_{j=1} \in \mathcal{X}^m$ collected from some data-generating process. Specifically, we are interested in approximating the posterior with density $\pi(\theta \mid x^o_{1:m}) \propto \prod^m_{j=1} p(x^o_j \mid \theta)\pi(\theta)$, where $\pi(\theta)$ is the prior density. As introduced below, this can be achieved via a neural SBI method which approximates the likelihood or posterior.

**Neural likelihood estimation (NLE).** NLEs [Papamakarios et al., 2019, Lueckmann et al., 2019, Boelts et al., 2022, Radev et al., 2023a] are extensions of the synthetic likelihood approach [Wood, 2010, Price et al., 2018] that use flexible conditional density estimators, typically normalising flows

[Rezende and Mohamed, 2015, Papamakarios et al., 2021], as surrogate models for the likelihood function associated with $\mathbb{P}_\theta$. The surrogate conditional density $q_\phi^{\text{NLE}} : \mathcal{X} \times \Theta \to [0, \infty)$ where $q_\phi^{\text{NLE}}(\cdot \mid \theta)$ is a density function for each $\theta \in \Theta$ and $\phi \in \Phi \subseteq \mathbb{R}^{d_\Phi}$ denotes its learnable parameters, is trained by minimising the negative log-likelihood with respect to $\phi$ on simulated samples. More precisely, let $\{(\theta_i, x_i)\}_{i=1}^n$ be the training data such that $\theta_i \sim \pi$ are realisations from the prior and $x_i \sim \mathbb{P}_{\theta_i}$ are realisations from the simulator. Then $\hat{\phi}_{\text{MC}} := \arg\min_{\phi \in \Phi} \ell_{\text{MC}}^{\text{NLE}}(\phi)$, where $\ell_{\text{MC}}^{\text{NLE}}$ is an empirical (Monte Carlo) estimate of negative expected log-density:

$$\ell^{\text{NLE}}(\phi) := -\mathbb{E}_{\theta \sim \pi}\left[\mathbb{E}_{x \sim \mathbb{P}_\theta}\left[\log q_\phi^{\text{NLE}}(x \mid \theta)\right]\right] \approx -\tfrac{1}{n}\sum_{i=1}^n \log q_\phi^{\text{NLE}}(x_i \mid \theta_i) =: \ell_{\text{MC}}^{\text{NLE}}(\phi).$$

Once the surrogate likelihood is trained, Markov chain Monte Carlo (MCMC) or variational inference methods are used to sample from the (approximate) posterior distribution $\pi_{\text{NLE}}(\theta \mid x_{1:m}^o) \propto \prod_{j=1}^m q_{\hat{\phi}_{\text{MC}}}^{\text{NLE}}(x_j^o \mid \theta)\pi(\theta)$. NLEs can therefore be regarded as being partially amortised—the surrogate likelihood need not be trained for a new observed dataset, however, MCMC needs to be carried out again to obtain the new posterior.

For a computationally costly simulator, we note that obtaining training samples can become a bottleneck, which affects the accuracy of estimating the expected loss. Thus, estimating the loss accurately with fewer samples is key to handling costly simulators.

**Neural posterior estimation (NPE).**   Instead of learning a surrogate likelihood, NPEs learn a mapping $x \mapsto p(\theta \mid x)$ from the data to the posterior using conditional density estimators. These are often based on mixture density networks [Bishop, 1994, Papamakarios and Murray, 2016] or normalising flows [Dinh et al., 2014, Papamakarios et al., 2017, Radev et al., 2022]. Similar to NLE, the conditional density $q_\phi^{\text{NPE}} : \Theta \times \mathcal{X}^m \to [0, \infty)$ is trained by minimising the negative log likelihood with respect to $\phi$ using data $\{(\theta_i, x_{1:m,i})\}_{i=1}^n$ generated by first sampling from the prior $\theta_i \sim \pi$ and then the simulator $x_{1:m,i} = (x_{1,i}, \ldots, x_{m,i}) \sim \mathbb{P}_{\theta_i}$:

$$\ell^{\text{NPE}}(\phi) := -\mathbb{E}_{\theta \sim \pi}\left[\mathbb{E}_{x_{1:m} \sim \mathbb{P}_\theta}\left[\log q_\phi^{\text{NPE}}(\theta \mid x_{1:m})\right]\right] \approx -\tfrac{1}{n}\sum_{i=1}^n \log q_\phi^{\text{NPE}}(\theta_i \mid x_{1:m,i}) =: \ell_{\text{MC}}^{\text{NPE}}(\phi)$$

Once $\phi$ is estimated, the NPE posterior is obtained as $\pi_{\text{NPE}}(\theta \mid x_{1:m}^o) = q_{\hat{\phi}_{\text{MC}}}^{\text{NPE}}(\theta \mid x_{1:m}^o)$. Although training $q_\phi^{\text{NPE}}$ incurs an upfront cost, this is a one-time cost as approximate posteriors for new observed datasets are obtained by a simple forward pass of $x_{1:m}^o$ through the trained networks, making NPEs fully amortised (in contrast with the partial amortisation of NLE). Similarly to NLE, the computationally costly step in NPE is the generation of training samples from running expensive simulators. Note that both $q_\phi^{\text{NLE}}$ and $q_\phi^{\text{NPE}}$ usually include a summary function (often architecturally implicit in NPE). This is helpful when $\mathcal{X}$ is high-dimensional or the number of observations $m$ is large [Alsing et al., 2018, Radev et al., 2022], and for NPE it allows conditioning on datasets of different sizes. Recently, alternative training objectives for NPE based on flow matching [Wildberger et al., 2023], diffusion [Geffner et al., 2023, Sharrock et al., 2024, Gloeckler et al., 2024], and consistency models [Schmitt et al., 2024b] have been proposed.

**Related work.**   We briefly note that beyond the aforementioned multi-fidelity methods, other works also aim to reduce the computational cost of SBI. In the context of ABC, adaptive sampling of the posterior using either sequential Monte Carlo techniques [Sisson et al., 2007, Beaumont et al., 2009, D. Moral et al., 2011] or Gaussian process surrogates [Gutmann and Corander, 2016, Meeds and Welling, 2014] has been a popular approach. A similar approach has been applied to neural SBI [Papamakarios and Murray, 2016, Lueckmann et al., 2017, Greenberg et al., 2019, Papamakarios et al., 2019, Hermans et al., 2020, Durkan et al., 2020], where sequential training schemes are employed to reduce the number of calls to the simulator. Other works have tackled this problem using cost-aware sampling [Bharti et al., 2025], side-stepping high-dimensional estimation [Jeffrey and Wandelt, 2020], early stopping of simulations [Prangle, 2016], dependent simulations [Niu et al., 2023, Bharti et al., 2023], expert-in-the-loop methods [Bharti et al., 2022b], self-consistency properties [Schmitt et al., 2024a], parallelisation of computations [Kulkarni and Moritz, 2023], and the Markovian structure of certain simulators [Gloeckler et al., 2025]. Our proposed method, introduced in Section 3, can be combined with all of these compute-efficient SBI methods and is hence complementary to them.

## 2.2 Multilevel Monte Carlo method

Consider some square-integrable function $f : \mathcal{Z} \to \mathbb{R}$ and distribution $\mu$ on a domain $\mathcal{Z} \subseteq \mathbb{R}^{d_{\mathcal{Z}}}$. We consider the task of estimating $\mathbb{E}_{z \sim \mu}[f(z)]$. A first approach is *standard Monte Carlo (MC)* [Robert and Casella, 2000, Owen, 2013], which yields the following estimator: $1/n \sum_{i=1}^{n} f(z_i)$, where $z_1, \ldots, z_n \sim \mu$. The root-mean-squared error (RMSE) of this estimator converges at a rate $O(n^{-1/2})$, where the rate constant is controlled by $\mathrm{Var}[f(z)]$ [Owen, 2013, Ch. 2]. In cases where $f$ is expensive to evaluate or $\mu$ is expensive to sample from, the RMSE can therefore be relatively large.

This issue can be mitigated by *multilevel Monte Carlo (MLMC)*, which was first proposed by Heinrich [2001], Giles [2008], and more recently reviewed in Giles [2015], Jasra et al. [2020]. Suppose we have a sequence of square-integrable functions $f_l : \mathcal{Z} \to \mathbb{R}$ for $l \in \{0, 1, \ldots, L\}$ which are approximations of $f$ and which are ordered such that $f_L = f$, and both the cost of evaluation $C_l$ and the accuracy (or *fidelity*) of $f_l$ increase with $l$. In that case, MLMC consists of expressing $\mathbb{E}_{z \sim \mu}[f(z)]$ through a telescoping sum and approximating each term through MC based on samples $z_i^l \sim \mu$ for $i = 1, \ldots, n_l$ and $l \in \{0, 1, \ldots, L\}$:

$$\mathbb{E}_{z \sim \mu}[f(z)] = \mathbb{E}_{z \sim \mu}[f_0(z)] + \sum_{l=1}^{L} \mathbb{E}_{z \sim \mu}[f_l(z) - f_{l-1}(z)]$$
$$\approx \frac{1}{n_0} \sum_{i=1}^{n_0} f_0(z_i^0) + \sum_{l=1}^{L} \left( \frac{1}{n_l} \sum_{i=1}^{n_l} \left( f_l(z_i^l) - f_{l-1}(z_i^l) \right) \right).$$

Note that this can also be thought of as using the low-fidelity functions as approximate control variates. By carefully balancing the number of samples $n_0, \ldots, n_L$ according to the costs $C_0, \ldots, C_L$ and variances $\mathrm{Var}[f_0(z)], \mathrm{Var}[f_1(z) - f_0(z)], \ldots, \mathrm{Var}[f_L(z) - f_{L-1}(z)]$ at each level, one can show that MLMC can significantly improve on the accuracy of MC given a fixed computational budget. For this reason, MLMC has found numerous applications in statistics and machine learning, including for optimisation [Asi et al., 2021, Hu et al., 2023, Yang et al., 2024], sampling [Dodwell et al., 2019, Jasra et al., 2020], variational inference [Fujisawa and Sato, 2021, Shi and Cornish, 2021], probabilistic numerics [Li et al., 2023, Chen et al., 2025] and the design of experiments [Goda et al., 2020, 2022].

## 3 Methodology

We now present NLE and NPE versions of our approach, termed *multilevel-NLE* and *multilevel-NPE*.

**MLMC for NLE and NPE.** Recall that we are performing inference for a simulator $(G_\theta, \mathbb{U})$ with a prior $\pi$ on the parameter $\theta \in \Theta$. We reparameterise the NLE and the NPE objective in terms of $u$ instead of $x$ (akin to the reparametrisation trick in variational inference), and express them more broadly using an arbitrary loss $\ell : \Phi \to \mathbb{R}$ (to represent either $\ell^{\text{NLE}}$ or $\ell^{\text{NPE}}$) and an arbitrary function $f_\phi : \mathcal{U}^m \times \Theta \to \mathbb{R}$ (to represent either $f_\phi^{\text{NLE}}$ or $f_\phi^{\text{NPE}}$) as:

$$\ell(\phi) := \mathbb{E}_{\theta \sim \pi, u_{1:m} \sim \mathbb{U}}[f_\phi(u_{1:m}, \theta)],$$

where for NLE we have $f_\phi^{\text{NLE}}(u, \theta) := -\log q_\phi^{\text{NLE}}(G_\theta(u) \mid \theta) = -\log q_\phi^{\text{NLE}}(x \mid \theta)$ with $m = 1$, and for NPE we have $f_\phi^{\text{NPE}}(u_{1:m}, \theta) := -\log q_\phi^{\text{NPE}}(\theta \mid G_\theta(u_1), \ldots, G_\theta(u_m)) = -\log q_\phi^{\text{NPE}}(\theta \mid x_{1:m})$. Hereafter, we present our methods using $f_\phi$ and $\ell$ in order to avoid duplication.

The MC estimator of the loss $\ell(\phi)$ is given by $\ell_{\text{MC}}(\phi) := \frac{1}{n} \sum_{i=1}^{n} f_\phi(u_{1:m,i}, \theta_i)$. Note that the variance of this MC estimator depends on the number of iid samples $n$. Hence, a small $n$ owing to a computationally expensive simulator will lead to a poor estimator for the loss. Now suppose that we have access to a sequence of generators $G_\theta^0(u), \ldots, G_\theta^{L-1}(u)$ with varying fidelity levels. Then, for $l = 0, \ldots, L - 1$, we can define a corresponding sequence of functions $f_\phi^l : \mathcal{U}^m \times \Theta \to \mathbb{R}$:

$$f_\phi^{\text{NLE},l}(u_{1:m}, \theta) := -\log q_\phi^{\text{NLE}}(G_\theta^l(u) \mid \theta), \text{ and } f_\phi^{\text{NPE},l}(u_{1:m}, \theta) := -\log q_\phi^{\text{NPE}}(\theta \mid G_\theta^l(u_1), \ldots, G_\theta^l(u_m)),$$

such that $G_\theta^L(u) := G_\theta(u)$ and $f_\phi^L(u_{1:m}, \theta) := f_\phi(u_{1:m}, \theta)$. This sequence of functions gives evaluations of the log conditional density at evaluations of the (approximate) simulator. Recall that the larger the value of $l$, the more accurate (and computationally expensive) such simulations will tend to be. At this point, we can re-express the objective using a telescoping sum as

$$\ell(\phi) := \mathbb{E}_{\theta \sim \pi, u_{1:m} \sim \mathbb{U}}[f_\phi(u_{1:m}, \theta)] = \mathbb{E}_{\theta \sim \pi, u_{1:m} \sim \mathbb{U}}\left[f_\phi^L(u_{1:m}, \theta)\right]$$
$$= \mathbb{E}_{\theta \sim \pi, u_{1:m} \sim \mathbb{U}}\left[f_\phi^0(u_{1:m}, \theta)\right] + \sum_{l=1}^{L} \mathbb{E}_{\theta \sim \pi, u_{1:m} \sim \mathbb{U}}\left[f_\phi^l(u_{1:m}, \theta) - f_\phi^{l-1}(u_{1:m}, \theta)\right]. \quad (1)$$

Equation (1) follows by adding then subtracting some terms, and using linearity of expectations. Now suppose that we can simulate from each of these approximate simulators to obtain:

$$\left\{\theta_i^l, u_{1:m,i}^l, \left\{G_{\theta_i^l}^l\left(u_{j,i}^l\right), G_{\theta_i^l}^{l-1}\left(u_{j,i}^l\right)\right\}_{j=1}^m\right\} \quad \text{where} \quad \theta_i^l \sim \pi, u_{1:m,i}^l \sim \mathbb{U},$$

for $i = 1, \ldots, n_l$ and $l = 0, \ldots, L$. We can then use these simulations to approximate each term in the telescoping sum through a Monte Carlo estimator as follows:

$$\ell(\phi) \approx \ell_{\text{MLMC}}(\phi) := \underbrace{\frac{1}{n_0} \sum_{i=1}^{n_0} f_\phi^0(u_{1:m,i}^0, \theta_i^0)}_{h_0(\phi)} + \sum_{l=1}^{L} \underbrace{\frac{1}{n_l} \sum_{i=1}^{n_l} \left(f_\phi^l(u_{1:m,i}^l, \theta_i^l) - f_\phi^{l-1}(u_{1:m,i}^l, \theta_i^l)\right)}_{h_l(\phi)} \quad (2)$$

This corresponds to an (unbiased) MLMC estimator of our original objective in (1). The first term, $h_0(\phi)$, approximates $\ell(\phi)$, but is biased since it uses $f_\phi^0$ (i.e. the lowest fidelity simulator) rather than $f_\phi^L$ (the highest fidelity simulator). The additional terms, $h_1(\phi), \ldots, h_L(\phi)$, correct this bias by estimating the expected difference between the objectives at consecutive fidelity levels.

Within each term $h_l(\phi)$, the functions $f_\phi^l$ and $f_\phi^{l-1}$ are evaluated on the same samples $u_{1:m}^l$ and $\theta^l$, i.e., they are *seed-matched*. This ensures that $f_\phi^l(u_{1:m,i}^l, \theta_i^l)$ and $f_\phi^{l-1}(u_{1:m,i}^l, \theta_i^l)$ are coupled and highly correlated, which leads to a reduction in variance [Owen, 2013, Ch. 8] since

$$\text{Var}[h_l(\phi)] \quad (3)$$
$$= \frac{1}{n_l}\left(\text{Var}[f_\phi^l(u_{1:m,i}^l, \theta_i^l)] + \text{Var}[f_\phi^{l-1}(u_{1:m,i}^l, \theta_i^l)] - 2\text{Cov}\left[f_\phi^l(u_{1:m,i}^l, \theta_i^l), \ f_\phi^{l-1}(u_{1:m,i}^l, \theta_i^l)\right]\right),$$

and the covariance will be large. This covariance will be particularly large the more similar the two functions $f_\phi^l$ and $f_\phi^{l-1}$ are, and we therefore expect $\text{Var}[h_l(\phi)]$ to be smallest in those settings. We can also immediately see that without seed-matching, the covariance will be small and the variance will be large, highlighting why seed-matching is essential for MLMC.

**Computational cost.** Let $C_l$ be the computational cost of sampling one $x$ from the $l^{\text{th}}$ level generator $G_\theta^l$ and evaluating $f_\phi^l$, and recall that $C_0 < C_1 < \ldots < C_L$. Then, the cost of MLMC is

$$\text{Cost}(\ell_{\text{MLMC}}(\phi); n_0, \ldots, n_L) = \mathcal{O}\left(n_0 C_0 + \sum_{l=1}^{L} n_l (C_l + C_{l-1})\right),$$

while that of the MC estimator is $\text{Cost}(\ell_{\text{MC}}(\phi); n) = \mathcal{O}(nC_L)$. Using solely the high-fidelity generator (as is customary in SBI) would require a large $n$ in order to reasonably estimate $\theta$, thus increasing the total cost. However, with multiple lower-fidelity generators available, we can have a different number of simulated samples per level (i.e. we can take $n_0 \neq n_1 \neq \ldots \neq n_L$), and can select $n_0, \ldots, n_L$ such that $n_l < n_{l-1}$. This allows us to take a much larger number of samples from the cheaper (or low $C_l$) approximations of the simulator, and a much smaller number of samples from the expensive (or high $C_l$) approximations of the simulator, making MLMC particularly attractive for reducing the total computational cost of simulation in neural SBI.

**Extensions.** There are several straightforward extensions of our approach which are not covered above so as to not overload notation. Firstly, each simulator could have its own base measure $\mathbb{U}^l$, which could be defined on spaces $\{\mathcal{U}^l\}_{l=1}^L$ of different dimensions. This is not a problem since we could simply consider $\mathbb{U}$ to be the tensor product measure and $\mathcal{U}$ to be the corresponding tensor product space, in which case all equations above remain valid. Similarly, the parameter space may differ across simulators. However, to ensure the best possible performance, it will still be essential to seed-match random numbers where there is overlap; see Owen [2013, Ch. 8] for more details on the use of common random numbers, and Section 5 for a study of this issue for multilevel neural SBI.

Secondly, although our discussion has pertained to multilevel-NLE and multilevel-NPE so far, it is straightforward to extend the MLMC approach to other neural SBI methods such as neural ratio estimation [Hermans et al., 2020, Durkan et al., 2020, Miller et al., 2022], score-based NPE [Geffner et al., 2023], flow-matching NPE [Wildberger et al., 2023], and GAN-based NPE [Ramesh et al., 2022] since these are all based on objectives which can be expressed as MC estimators.

**Optimisation.** Gradient-based optimisation of the MLMC objective $\ell_{\text{MLMC}}(\phi)$ can be challenging due to the 'conflicting' gradients which appear in consecutive terms $\nabla_\phi h_l(\phi)$ and $\nabla_\phi h_{l+1}(\phi)$. More precisely, the term $\nabla_\phi h_l(\phi)$ always contains

$$\zeta_\phi^{l,+} := \frac{1}{n_l} \sum_{i=1}^{n_l} \nabla_\phi f_\phi^l(u_{1:m,i}^l, \theta_i^l),$$

which approximates $\nabla_\phi \mathbb{E}[f_\phi^l]$, whilst the term $\nabla_\phi h_{l+1}(\phi)$ always contains

$$\zeta_\phi^{l,-} := -\frac{1}{n_{l+1}} \sum_{i=1}^{n_{l+1}} \nabla_\phi f_\phi^l(u_{1:m,i}^{l+1}, \theta_i^{l+1}),$$

which approximates $-\nabla_\phi \mathbb{E}[f_\phi^l]$. In the infinite-sample limit, $\nabla_\phi \mathbb{E}[f_\phi^l]$ and $-\nabla_\phi \mathbb{E}[f_\phi^l]$ cancel out, but this is not the case for $\zeta_\phi^{l,+}$ and $\zeta_\phi^{l,-}$ since we are typically working with only a small number of expensive

---

**Algorithm 1** MLMC gradient adjustment

**Input** $\nabla_\phi h_0(\phi), \{\zeta_\phi^{l,+}, \zeta_\phi^{l-1,-}\}_{l=1}^L, \epsilon > 0 \approx 0$
// *Rescaling the gradients:*
**for** $l = 1$ to $L$
$\qquad \zeta_\phi^{l-1,-} \leftarrow \frac{\|\zeta_\phi^{l,+}\|_2}{\|\zeta_\phi^{l-1,-}\|_2 + \epsilon} \zeta_\phi^{l-1,-}$
**end for**
$\nabla_\phi h_c(\phi) \leftarrow \sum_{l=1}^L (\zeta_\phi^{l,+} + \zeta_\phi^{l-1,-})$
// *Projecting the gradients (only when conflicting)*
**if** $\nabla_\phi h_0(\phi) \cdot \nabla_\phi h_c(\phi) < 0$ **then**
$\qquad \tilde{\nabla}_\phi h_0(\phi) \leftarrow \nabla_\phi h_0(\phi) - \frac{\nabla_\phi h_0(\phi) \cdot \nabla_\phi h_c(\phi)}{\|\nabla_\phi h_c(\phi)\|_2^2} \nabla_\phi h_c(\phi)$
$\qquad \tilde{\nabla}_\phi h_c(\phi) \leftarrow \nabla_\phi h_c(\phi) - \frac{\nabla_\phi h_0(\phi) \cdot \nabla_\phi h_c(\phi)}{\|\nabla_\phi h_0(\phi)\|_2^2} \nabla_\phi h_0(\phi)$
**else** $\tilde{\nabla}_\phi h_0(\phi) \leftarrow \nabla_\phi h_0(\phi), \tilde{\nabla}_\phi h_c(\phi) \leftarrow \nabla_\phi h_c(\phi)$
**end if**
**Output** $\tilde{\nabla}_\phi \ell_{\text{MLMC}}(\phi) \leftarrow \tilde{\nabla}_\phi h_0(\phi) + \tilde{\nabla}_\phi h_c(\phi)$

---

simulations. When naively applying standard gradient-based optimisation methods on this loss, we observe that the training dynamics is typically dominated by only one of these two quantities until approaching stationarity, at which point the conflicting gradients lead to unstable updates and, ultimately, often cause divergence.

To mitigate this issue, we use a combination of gradient adjustments summarised in Algorithm 1. Firstly, we rescale $\zeta_\phi^{l,+}$ and $\zeta_\phi^{l,-}$ to ensure that they have comparable norms and that their difference remains small and stable. Secondly, we apply the gradient projection technique of Liu et al. [2020], projecting the gradients of $h_0(\phi)$ and $h_c(\phi) := \sum_{l=1}^L h_l(\phi)$ onto each other's normal planes to reduce the impact of conflicting gradients. We observe empirically that combining these two techniques significantly improves the stability of the optimisation throughout the training and leads to better performance; see Appendix B.6 for a detailed comparison.

## 4 Theory

We now present our main theoretical results. Theorem 1 expresses the variance of each of the terms in the telescoping sum approximation (see (2)) as a function of the number of simulations per level, and the magnitude of the difference in generators between consecutive levels.

We say that $\mu$ is log-concave if it has a density of the form $\exp(-\psi(z))$ for some convex function $\psi : \mathcal{Z} \to \mathbb{R}$. We recall that for $r \in [1, \infty)$, $d, d' \in \mathbb{N}$ and a non-empty, open, connected set $\mathcal{Z} \subseteq \mathbb{R}^d$, the space of vector-valued $r$-integrable functions with respect to a probability distribution $\mu$ is given by $L^r(\mu) := \{g : \mathcal{Z} \to \mathbb{R}^{d'} : \|g\|_{L^r(\mu)} := (\int_{\mathcal{Z}} \|g(x)\|_2^r \mu(dx))^{1/r} < \infty\}$. For $\tau \in \mathbb{N}$, the corresponding Sobolev space of vector-valued functions of smoothness $\tau$ is given by $W^{\tau,r}(\mu) := \{g : \mathcal{Z} \to \mathbb{R}^{d'} : \|g\|_{W^{\tau,r}(\mu)} = (\sum_{|\alpha| \leq \tau} \|D^\alpha g\|_{L^r(\mu)}^r)^{1/r} < \infty\}$, where for a multi-index $\alpha \in \mathbb{N}^d$, $D^\alpha$ is the weak derivative operator corresponding to $\alpha$. Finally, we recall that a function $g$ is locally $K_{\text{Lip}}$-smooth if its gradient is locally Lipschitz continuous with Lipschitz constant $K_{\text{Lip}} > 0$; i.e. for all $z, z'$ in some open set of $\mathcal{Z}$, we have that $\|\nabla g(z) - \nabla g(z')\|_2 \leq K_{\text{Lip}}\|z - z'\|_2$. For simplicity, we will write $\tilde{q}_\phi(x_{1:m}, \theta)$ for the conditional density model used for either NLE (in which case $\tilde{q}_\phi(x_{1:m}, \theta) = q_\phi^{\text{NLE}}(x_1|\theta)$) and NPE (in which case $\tilde{q}_\phi(x_{1:m}, \theta) = q_\phi^{\text{NPE}}(\theta|x_{1:m})$).

**Theorem 1.** *Let $\phi \in \Phi$ and suppose the following assumptions hold:*

- *(A1)* *The prior $\pi$ and the base measure $\mathbb{U}$ are log-concave distributions.*

- *(A2)* *The generators satisfy $\|G^l\|_{W^{1,4}(\pi \times \mathbb{U})} \leq S_l$ for $l \in \{0, 1, \ldots, L\}$.*

- *(A3)* *$\log \tilde{q}_\phi$ is continuously differentiable, locally $K_{\text{Lip}}(\phi)$−smooth and satisfies the growth condition $\|\nabla \log \tilde{q}_\phi(x_{1:m}, \theta)\|_2 \leq K_{\text{grow}}(\phi)(\sum_{i=1}^m \|x_i\|_2 + \|\theta\|_2 + 1)$ for some $K_{\text{Lip}}(\phi), K_{\text{grow}}(\phi) > 0$.*

*Then, for $l \in \{1, \ldots, L\}$ and $K_0(\phi), \ldots, K_L(\phi) > 0$ independent of $n_0, \ldots, n_L$, we have that:*

$$Var\left[h_0(\phi)\right] \leq \frac{K_0(\phi)}{n_0} \left( \left\| G^0 \right\|_{W^{1,4}(\pi \times \mathbb{U})}^4 + 1 \right),$$

$$and \; Var\left[h_l(\phi)\right] \leq \frac{K_l(\phi)}{n_l} \left\| G^l - G^{l-1} \right\|_{W^{1,4}(\pi \times \mathbb{U})}^2.$$

See Appendix A.1 for the proof, and we emphasise that the variance is over both parameters $\theta$ and noise, but conditional on $\phi$. (A1) requires log-concavity, which is a strong condition, but this is only required of the prior and the base measure. However, in SBI these tend to be simple distributions such as Gaussians or uniforms, which satisfy this assumption; see Saumard and Wellner [2014] for how to verify log-concavity in practice. (A2) is relatively mild: it asks that $G^0, \ldots, G^L$ have at least one derivative in $\theta$ and $u$, and for these generators and their derivatives to have a fourth moment. This will hold when the simulators are used to define sufficiently well-behaved data-generating processes, but will be violated for sufficiently heavy-tailed distributions (e.g. the Cauchy). Finally, (A3) is mild; it holds when the gradient of the log conditional density estimator is Lipschitz continuous, which is for example the case when the conditional density is twice continuously differentiable and the Hessian is bounded (see e.g. Lemma 2.3 of Wright and Recht [2022]). It also holds for models such as mixtures of Gaussians, which are used in mixture density networks and for normalising flows with sufficiently regular transformations; see Table 1 in Liang et al. [2022] for some known Lipschitz continuous transformations. There are two key implications of Theorem 1. The first is a bound on the variance of the MC objective:

$$\mathrm{Var}\left[\ell_{\mathrm{MC}}(\phi)\right] \leq \frac{K(\phi)}{n} \left( \left\| G \right\|_{W^{1,4}(\pi \times \mathbb{U})}^4 + 1 \right), \tag{4}$$

which is obtained by noticing that the bound on $\mathrm{Var}[h_0(\phi)]$ is simply a bound on a Monte Carlo objective. From this, we immediately notice that the bound will be large whenever the high-fidelity simulator is expensive and $n$ is small, or whenever the simulator is complex as measured in this Sobolev norm. The second implication is a bound on the variance of the MLMC objective:

$$\mathrm{Var}\left[\ell_{\mathrm{MLMC}}(\phi)\right]$$
$$\leq \frac{K_0(\phi)}{n_0} \left( \left\| G^0 \right\|_{W^{1,4}(\pi \times \mathbb{U})}^4 + 1 \right) + \sum_{l=1}^{L} \frac{K_l(\phi)}{n_l} \left\| G^l - G^{l-1} \right\|_{W^{1,4}(\pi \times \mathbb{U})}^2. \tag{5}$$

In order to make this bound small, we need to make each term small. Typically, $\|G^0\|_{W^{1,4}(\pi \times \mathbb{U})}$ will be large, but this will be counterbalanced by taking $n_0$ to be large. For the higher-fidelity levels, the number of samples $n_l$ will typically be smaller, but this will be counter-balanced by the fact that $\|G^l - G^{l-1}\|_{W^{1,4}(\pi \times \mathbb{U})}$ is small whenever $G^{l-1}$ is a good approximation of $G^l$. As we now show, we can directly use (5) to get an indication of how to select $n_0, \ldots, n_L$.

**Theorem 2.** *Suppose the assumptions of Theorem 1 hold. Then, the values of $n_0, \ldots, n_L$ which minimise the upper bound on $Var[\ell_{MLMC}(\phi)]$ in (5) given a fixed computational budget; i.e. for $Cost(\ell_{MLMC}(\phi); n_0, \ldots, n_L) \leq C_{budget}$ for some $C_{budget} > 0$, are given by*

$$n_0^\star \propto \frac{C_{budget}}{\sqrt{C_0}} \sqrt{\left\| G^0 \right\|_{W^{1,4}(\pi \times \mathbb{U})}^4 + 1}, \quad n_l^\star \propto \frac{C_{budget}}{\sqrt{C_l + C_{l-1}}} \left\| G^l - G^{l-1} \right\|_{W^{1,4}(\pi \times \mathbb{U})}.$$

See Appendix A.2 for the proof. This result provides useful intuition on how to select the number of samples per level. For instance, consider the case $L = 1$, i.e., two levels. If $G^0$ is known to be a good approximation of $G^1$, it makes sense to allocate a large budget to generating low-fidelity simulations while only a small number of high-fidelity simulations would be sufficient. On the other hand, if $G^0$ and $G^1$ differ substantially, allocating a larger budget to high-fidelity simulations makes sense, despite the higher cost, as it can help accurately capture this difference. Although Theorem 2 provides intuition, it may be hard to obtain the optimal $n_0, \ldots, n_L$ exactly since it requires computing quantities which are often unknown. For example, computing Sobolev norms can be challenging, and the results implicitly depend on $\phi$ through the constants $K_0(\phi), \ldots, K_L(\phi)$ of Theorem 1 (which are unlikely to be tight). This is a common limitation of MLMC theory; see [Giles, 2015, Sec. 2-3].

Before concluding this section, we note that our theory focussed on the variance of $\ell_{\mathrm{MLMC}}(\phi)$, but another important quantity for gradient-based optimisation will be the variance of the gradient $\nabla_\phi \ell_{\mathrm{MLMC}}(\phi)$ of this objective. It turns out that similar results are straightforward to prove for this quantity under very minor modifications of the assumptions, see Appendix A.3.

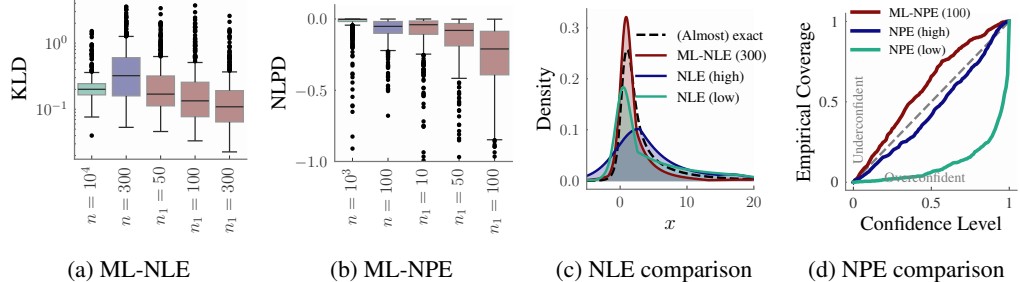

| (a) ML-NLE | (b) ML-NPE | (c) NLE comparison | (d) NPE comparison |

Figure 2: Performance of our ML-NLE and ML-NPE method on the g-and-k example. (a) KL-divergence ($\downarrow$) between the estimated and the (almost) exact density for ML-NLE under different high-fidelity samples $n_1$. We compare it with NLE (low) trained on only low-fidelity data ($n = 10^4$) and NLE (high), trained on only high-fidelity data ($n = 300$). (b) Negative log-posterior density (NLPD $\downarrow$) for ML-NPE, NPE (low) with $n = 10^3$, and NPE (high) with $n = 100$. (c) One instance of learned densities using NLE. (d) Empirical coverage plot for ML-NPE, NPE (low), and NPE (high).

## 5 Numerical Experiments

We compare the performance of our multilevel version of NLE and NPE, termed ML-NLE and ML-NPE, respectively, against their standard counterpart with the MC loss. We use the sbi library [Tejero-Cantero et al., 2020] implementation for NLE and NPE , see Appendix B for the details. The code to reproduce our experiments is available at https://github.com/yugahikida/multilevel-sbi.

### 5.1 The g-and-k distribution: an illustrative example

We first consider the g-and-k distribution [Prangle, 2020] as an illustrative example. This is a very flexible univariate distribution that is defined via its quantile function and has four parameters, controlling the mean, variance, skewness, and kurtosis respectively, making it challenging for SBI methods. It does not typically have a low-fidelity simulator, so we construct one through a Taylor approximation of the quantile function, see Appendix B.1 for details. This makes it a slightly contrived example, but the fact that the g-and-k allows for an efficient approximation of the likelihood will make it particularly convenient to study the performance of NLE-based methods.

We fix the number of low-fidelity samples ($n_0 = 10^4$ for ML-NLE and $n_0 = 10^3$ for ML-NPE) and vary the number of high-fidelity samples $n_1$ to asses the improvement in performance of our methods as $n_1$ increases. For ML-NLE, we compute the Kullback-Leibler divergence (KLD) between the estimated conditional density and a numerical approximation of the likelihood. For ML-NPE, we use the negative log-posterior density (NLPD) of the true $\theta$ under the estimated posterior density as the metric. The results in Figure 2a and 2b show that the multilevel versions of NLE and NPE perform better than their standard counterparts (with MC loss) using just a fraction of the high-fidelity samples. Unsurprisingly, the performance of MLE-NLE and ML-NPE improves as $n_1$ increases.

In Figure 2c, we show an example of the NLE densities learned, with additional examples in Appendix B.1. Our ML-NLE method is able to approximate the almost exact g-and-k density the best. The coverage plot in Figure 2d shows that ML-NPE yields slightly conservative posteriors as opposed to the overconfident posteriors obtained from NPE trained on either all low- or all high-fidelity data.

### 5.2 Ornstein–Uhlenbeck process: a popular financial model

Our next experiment involves the Ornstein-Uhlenbeck (OU) process—a stochastic differential equation model commonly used in financial analysis [Minenna, 2003], which in our case outputs a 100-dimensional Markovian time-series and has three parameters. The process is known to converge to a stationary Gaussian distribution, which we use as the low-fidelity simulator, see Appendix B.3. The example is reproduced from Krouglova et al. [2025], who used it for benchmarking their transfer learning approach to NPE (TL-NPE). TL-NPE first trains an NPE network on low-fidelity simulations, and then uses a small set of high-fidelity data to refine the network parameters until a stopping criterion is met. We implement TL-NPE with $n_0 = 1100$ and $n_1 = \{10, 100\}$, and compare against ML-NPE with $n_0 = 1000$ and $n_1 = \{10, 100\}$. These values are picked to keep the simulation

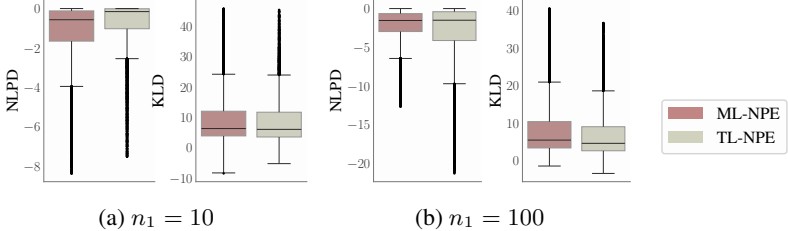

(a) $n_1 = 10$             (b) $n_1 = 100$

Figure 3: Performance of ML-NPE and TL-NPE measured by NLPD ($\downarrow$) and KL divergence ($\downarrow$) with different number of high-fidelity samples $n_1$ on the OU process. (a) When $n_1 = 10$, ML-NPE outperforms on both metric. (b) When $n_1 = 100$, TL-NPE outperforms on both metric. Note that for visualisation purpose, we removed outliers, see Appendix B.3 for the result with all the data.

budget the same for both methods. TL-NPE uses the default early stopping criterion from the `sbi` library [Tejero-Cantero et al., 2020], which terminates training if the validation loss increases for 20 epochs. Once the criterion is met, the network parameters achieving the lowest validation loss are selected. This criterion is used for both the low- and the high-fidelity training stages. We use 20% of the data as validation set for early stopping.

We run both methods 20 times and compute the NLPD of the ground-truth parameter $\theta$ under the estimated posterior across 500 test points. Additionally, we report the KLD between the posterior obtained using TL-NPE or ML-NPE and a reference NPE posterior trained with $n = 10{,}000$ high-fidelity simulations. Results are aggregated across the 20 runs and all test points, and reported in Figure 8. We observe that both the methods achieve comparable performance: for $n_1 = 10$, ML-NPE performs slightly better on both metrics, whereas for $n_1 = 100$, TL-NPE slightly outperforms ML-NPE on both.

We additionally investigate a challenging scenario where the high- and low-fidelity simulators have differing dimensionalities of $\theta$, in which our method exhibits some limitations, possibly due to instability during optimisation; see Appendix B.3.1 for details.

### 5.3 Toggle-switch model: a Systems Biology example

We now consider the toggle-switch model [Bonassi et al., 2011, Bonassi and West, 2015]. This model describes the interaction between two genes over time, and has seven parameters and a scalar observation at the end of a time interval. Simulators typically use time-discretisation, and the total number of time-steps $T$ acts as a fidelity parameter: running the model with large $T$ incurs larger computational cost but leads to accurate simulations, while smaller $T$ leads to cheap but inaccurate samples. Thus, this model illustrates a setting with more than two fidelity levels (by taking $T$ to be more than two values). We take $T_0 = 50$, $T_1 = 80$, and $T_2 = 300$ to be the number of steps for the three fidelity levels with $n_0 = 10^4$, $n_1 = 500$, and $n_2 = 100$. Guided by the intuition from Theorem 2, we allocate a large budget to estimating the difference between the low- and the medium-fidelity simulators, leveraging prior knowledge that this difference is substantial. See Appendix B.4.1 for results under alternative budget allocations. Note that in this case each fidelity level has a different base measure of dimension $2T+1$; however,

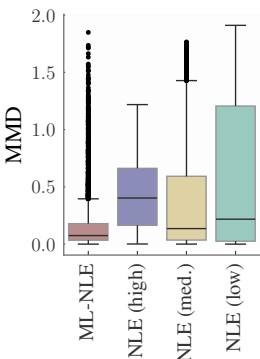

Figure 4: MMD ($\downarrow$) across 5000 parameter values for NLE and ML-NLE.

there is still sufficient seed-matching (see Appendix B.4 for details), and therefore variance reduction, thanks to MLMC. We compare our ML-NLE method with the standard NLE trained on samples from either the low- ($n = 12{,}060$), the medium- ($n = 7537$), or the high-fidelity simulator ($n = 2010$). The number of training data $n$ in each case is selected so as to match the total computational cost of simulation between ML-NLE and NLE, see Appendix B.4. Figure 4 reports the maximum mean discrepancy (MMD) [Gretton et al., 2012] between 500 samples from the learned conditional densities and 500 samples from the high-fidelity simulator across for 5000 different parameter values. We observe that ML-NLE performs better than all the NLE baselines for the same computational cost.

## 5.4  Cosmological Simulations

We now consider a cosmological simulator using the CAMELS suite [Villaescusa-Navarro et al., 2021, 2023]—one of the most computationally intensive cosmological simulations to date—which comprises both low- and high-fidelity data. These are state-of-the-art simulations being used with real-world data. Developing surrogate, multilevel techniques for cosmology has become a key research focus [Chartier et al., 2021, Chartier and Wandelt, 2022]. Our task is to infer a standard cosmological target parameter using a 39-dimensional power spectra of cosmology data, see Appendix B.5 for an example. The low-fidelity simulations are gravity-only N-body simulations, whose physical behaviour is controlled only by the parameter and some Gaussian fluctuations of the initial conditions. The high-fidelity hydrodynamic simulations have additional physics, controlled by an additional five parameters. We include these additional parameters as part of the $\mathcal{U}$ space, constituting a case of partial common random numbers between the low- and high-fidelity simulators, similar to Section 5.3.

Here, the high-fidelity simulations can be orders of magnitude ($> \times 100$) slower to generate than the low-fidelity ones, making this a representative problem for our method. Assuming we only have access to $n = n_1 = 20$ high-fidelity simulations, we wish to ascertain the improvement in inference accuracy by including 1000 low-fidelity simulations (i.e. $n_0 = 980$) using our ML-NPE method. To that end, we measure the NLPD and empirical coverage of the estimated posteriors for 980 test data. The result in Figure 5 shows that ML-NPE performs better than standard NPE for both the metrics. Standard NPE tends to produce overconfident posteriors, while ML-NPE yields calibrated or underconfident posteriors for most confidence levels. Thus, including low-fidelity samples using our method leads to better inference outcomes. Before concluding, we note that some recent papers demonstrating the potential of multi-fidelity SBI methods in cosmology appeared around the same time as our paper [Saoulis et al., 2025, Thiele et al., 2025]. These more in-depth studies clearly highlight the potential for impact of advanced multi-fidelity SBI methods.

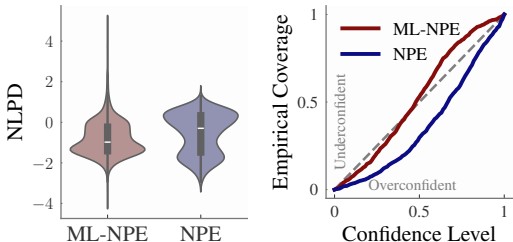

Figure 5: NLPD ↓ and empirical coverage of NPE and ML-NPE for the cosmological inference task.

## 6  Conclusion

This paper demonstrated how to reduce the cost of SBI using MLMC, but could more broadly be seen as a way to perform multilevel training of conditional density estimators in scenarios where data from different sources with different accuracy levels needs to be combined. Our method can be readily applied to scenarios with more than two fidelity levels. It is also particularly appealing since it is complementary to other compute-efficient SBI methods. For example, Tatsuoka et al. [2025] recently proposed to train an NPE network on low-fidelity data, and to then use the resulting posterior approximation to guide sampling from the high-fidelity simulator. This approach could easily be combined with our method.

In terms of limitations, our method involves gradient adjustments during optimisation, and we did observe a minor increase of roughly 15%-20% in training time compared to that of standard SBI methods (see Appendix B.2). However, this is not a significant issue as training time is usually negligible compared to costly high-fidelity simulations. Another limitation of our approach is that it is not applicable in cases where seed-matching of the low- and high-fidelity simulators is not possible. This was not an issue in any of the examples we encountered, but limit its applicability in some cases.

### Acknowledgments

The authors are grateful to Sam Power, Tim Sullivan and David Warne for helpful discussions, and to the authors of Krouglova et al. [2025] for sharing their code and identifying a bug in our implementation of their method in a preprint version of this paper. YH and AB were supported by the Research Council of Finland grant no. 362534. FXB was supported by the EPSRC grant [EP/Y022300/1]. NJ was supported by the ERC-selected UKRI Frontier Research Grant EP/Y03015X/1 and by the Simons Collaboration on Learning the Universe.

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

# Supplemental Material

The appendix is arranged as follows: Appendix A contains the proofs of the theoretical results presented in Section 4. Appendix B consists of the experimental details and additional results.

## A    Proof of theoretical results

### A.1    Proof of Theorem 1

*Proof.* Note that here, and throughout the rest of this proof, we will simplify the notation. We will drop the subscript for variances and expectations, and these should all be understood as being $u_i \sim \mathbb{U}$ for $i \in \{1, \ldots, m\}$ and $\theta \sim \pi$. We will also use the variable $z$ to denote a vector containing $u_{1:m}$ and $\theta$, so that $z \in \mathbb{R}^{md_u + d_\Theta}$. Finally, we will write $L^4$ and $W^{1,4}$ to denote the spaces $L^4(\pi \times \mathbb{U})$ and $W^{1,4}(\pi \times \mathbb{U})$.

The proof will be structured as follows. We will express the variance of each term using the variance of individual samples. We will then use a Poincaré-type inequality to bound the variance of the objective in terms of the expected squared norm of its gradient, then we will upper bound the norm using only terms which depend on constants and terms expressing how well $G^{l-1}$ approximates $G^l$.

Since each term in the MLMC expansion is an MC estimator (and therefore based on independent samples), we can express the variances of each term as follows:

$$\text{Var}\left[h_0(\phi)\right] = \text{Var}\left[\frac{1}{n_0} \sum_{i=1}^{n_0} f_\phi^0(z_i)\right]$$

$$= \frac{1}{n_0^2} \sum_{i=1}^{n_0} \text{Var}\left[f_\phi^0(z_i)\right] = \frac{1}{n_0} \text{Var}\left[f_\phi^0(z)\right], \tag{6}$$

where we use the fact that the variance of a sum of independent random variables is the sum of variances. Similarly for the other levels,

$$\text{Var}\left[h_l(\phi)\right] = \frac{1}{n_l} \text{Var}\left[f_\phi^l(z) - f_\phi^{l-1}(z)\right] \quad \text{for } l \in \{1, \ldots, L\}. \tag{7}$$

For the first step, we use a version of a Poincaré-type inequality for log-concave measures due to [Bobkov, 1999] (note that a simpler statement and additional discussion is provided in Proposition 10.1 (b) of Saumard and Wellner [2014] for the case of strongly log-concave measures). This result shows that for any log-concave measure $\mu$, there exists $K_{\text{Poin}} > 0$ such that for any sufficiently regular integrand $f : \mathcal{Z} \to \mathbb{R}$ where $\mathcal{Z} \subseteq \mathbb{R}^{d_z}$, $\text{Var}[f(z)] \leq K_{\text{Poin}} \mathbb{E}_{z \sim \mu}\left[\|\nabla f(z)\|_2^2\right]$. Applying this Poincaré inequality to each term of the learning objective (i.e. to the terms in (6) and (7)), we get

$$\text{Var}\left[f_\phi^0(z)\right] \leq K_{\text{Poin}} \, \mathbb{E}\left[\left\|\nabla_z f_\phi^0(z)\right\|_2^2\right] \tag{8}$$

$$\text{Var}\left[f_\phi^l(z) - f_\phi^{l-1}(z)\right] \leq K_{\text{Poin}} \, \mathbb{E}\left[\left\|\nabla_z f_\phi^l(z) - \nabla_z f_\phi^{l-1}(z)\right\|_2^2\right] \tag{9}$$

for $l \in \{1, \ldots, L\}$. Here, we emphasise again that the expectation is over $z$, which encompasses both $\theta$ and $u_{1:m}$, and the vector $\nabla_z f_\phi^l(z) \in \mathbb{R}^{d_u m + d_\Theta}$ for any $l \in \{1, \ldots, L\}$. This result requires the joint distribution of the prior $\pi$ and $m$ times the base measure $\mathbb{U}$ to be log-concave, which holds since the product of log-concave densities is also log-concave (since the sum of convex functions is convex) and we have assumed that the prior and base measures are independent and separately log-concave through Assumption (A1).

To simplify notation, we now introduce the vector-valued function $g^l(z) = g^l(\theta, u_{1:m}) = (G_\theta^l(u_1)^\top, \ldots, G_\theta^l(u_m)^\top, \theta^\top)^\top$ so that $\tilde{q}_\phi(x_{1:m}, \theta) = \tilde{q}_\phi(g^l(\theta, u)) = \tilde{q}_\phi(g^l(z))$. We will now derive our first bound, which looks at the first term in the MLMC expansion. To do so, we simplify

(8) as follows

$$\mathbb{E}\left[\left\|\nabla_z f_\phi^0(z)\right\|_2^2\right] = \mathbb{E}\left[\left\|\nabla_z - \log \tilde{q}_\phi\left(g^0(z)\right)\right\|_2^2\right]$$

$$= \mathbb{E}\left[\left\|\nabla_z \log \tilde{q}_\phi\left(g^0(z)\right)\right\|_2^2\right]$$

$$= \mathbb{E}\left[\left\|\nabla_z g^0(z)\nabla \log \tilde{q}_\phi\left(g^0(z)\right)\right\|_2^2\right] \tag{10}$$

$$\leq \mathbb{E}\left[\left\|\nabla_z g^0(z)\right\|_2^2\left\|\nabla \log \tilde{q}_\phi\left(g^0(z)\right)\right\|_2^2\right] \tag{11}$$

$$\leq \mathbb{E}\left[\left\|\nabla_z g^0(z)\right\|_2^4\right]^{\frac{1}{2}}\mathbb{E}\left[\left\|\nabla \log \tilde{q}_\phi\left(g^0(z)\right)\right\|_2^4\right]^{\frac{1}{2}} \tag{12}$$

Here, we have that (10) follows due to the chain rule, (11) follows due to the definition of the matrix 2-norm, (12) follows due to the Cauchy-Schwarz inequality of expectations.

We now bound each term in (12) separately. For the first term, we get

$$\|\nabla_z g^0(z)\|_2^2 \leq \sum_{i=1}^m \|\nabla_{u_i} g^0(z)\|_2^2 + \|\nabla_\theta g^0(z)\|_2^2 \tag{13}$$

$$\leq \sum_{i=1}^m \sum_{j=1}^m \|\nabla_{u_i} G_\theta^0(u_j)\|_2^2 + \sum_{j=1}^m \|\nabla_\theta G_\theta^0(u_j)\|_2^2 + \|\nabla_\theta \theta\|_2^2 \tag{14}$$

$$= \sum_{j=1}^m \|\nabla_{u_j} G_\theta^0(u_j)\|_2^2 + \sum_{j=1}^m \|\nabla_\theta G_\theta^0(u_j)\|_2^2 + d_\Theta \tag{15}$$

$$= \sum_{j=1}^m \sum_{k=1}^{d_\mathcal{U}} \|\nabla_{u_{jk}} G_\theta^0(u_j)\|_2^2 + \sum_{j=1}^m \sum_{k=1}^{d_\Theta} \|\nabla_{\theta_k} G_\theta^0(u_j)\|_2^2 + d_\Theta \tag{16}$$

where (13) and (14) follow from the fact that the two norm squared of a matrix is less than the sum of the two norm squared of sub-matrices constructed through rows and columns, (15) follows by noticing that $\|\nabla_\theta \theta\|_2^2 = d_\Theta$ and $\|\nabla_{u_i} G_\theta^0(u_j)\|_2^2 = 0$ whenever $i \neq j$, and (16) follows similarly to (13) and (14). Taking squares and an expectation, we get:

$$\mathbb{E}\left[\|\nabla_z g^0(z)\|_2^4\right]$$

$$\leq \mathbb{E}\left[\left(\sum_{j=1}^m \sum_{k=1}^{d_\mathcal{U}} \|\nabla_{u_{jk}} G_\theta^0(u_j)\|_2^2 + \sum_{j=1}^m \sum_{k=1}^{d_\Theta} \|\nabla_{\theta_k} G_\theta^0(u_j)\|_2^2 + d_\Theta\right)^2\right] \tag{17}$$

$$\leq (md_\mathcal{U} + md_\Theta + 1)\mathbb{E}\left[\sum_{j=1}^m \sum_{k=1}^{d_\mathcal{U}} \|\nabla_{u_{jk}} G_\theta^0(u_j)\|_2^4 + \sum_{j=1}^m \sum_{k=1}^{d_\Theta} \|\nabla_{\theta_k} G_\theta^0(u_j)\|_2^4 + d_\Theta^2\right] \tag{18}$$

$$\leq (md_\mathcal{U} + md_\Theta + 1)\left(m\|G^0\|_{W^{1,4}}^4 + d_\Theta^2\right) \tag{19}$$

$$\leq K_{\text{grad}}\left(\|G^0\|_{W^{1,4}}^4 + 1\right) \tag{20}$$

where (18) follows from $(\sum_{i=1}^n a_i)^2 \leq n\sum_{i=1}^n a_i^2$, (19) follows from the definition of the Sobolev norm and the fact that $u_1, \ldots, u_m$ have the same distribution and hence the same expectation, and (20) follows by grouping constants together.

We now move on to bounding the second term in (12). Since we assumed in Assumption (A3) that $\nabla \log \tilde{q}_\phi$ satisfies a linear growth condition, we must have that

$$\mathbb{E}\left[\|\nabla \log \tilde{q}_\phi(g^0(z))\|_2^4\right] \leq \mathbb{E}\left[\left(K_{\text{grow}}(\phi)\left(\sum_{j=1}^m \|G_\theta^0(u_j)\|_2 + \|\theta\|_2 + 1\right)\right)^4\right] \quad (21)$$

$$\leq (m+2)^3 K_{\text{grow}}(\phi)^4 \mathbb{E}\left[\sum_{j=1}^m \|G_\theta^0(u_j)\|_2^4 + \|\theta\|_2^4 + 1\right] \quad (22)$$

$$\leq (m+2)^3 K_{\text{grow}}(\phi)^4 \left(m\|G^0\|_{W^{1,4}}^4 + \mathbb{E}\left[\|\theta\|_2^4\right] + 1\right) \quad (23)$$

$$\leq K_{\text{score}}(\phi)\left(\|G^0\|_{W^{1,4}}^4 + 1\right) \quad (24)$$

Here, (21) uses the growth condition, (22) holds by applying $(\sum_{i=1}^n a_i)^4 \leq n^3 \sum_{i=1}^n a_i^4$, (23) follows from the definition of Sobolev norm, and (24) uses the fact that $\mathbb{E}\left[\|\theta\|_2^4\right]$ is upper bounded by a constant since the expectation is against $\pi$, which a log-concave distribution and hence all its moments are finite (see Section 5.1. of Saumard and Wellner [2014]).

Combining (6), (8), (12), (20), and (24) therefore gives:

$$\text{Var}\left[h_0(\phi)\right] \leq \frac{1}{n_0} K_{\text{Poin}} K_{\text{grad}}^{\frac{1}{2}} \left(\|G^0\|_{W^{1,4}}^4 + 1\right)^{\frac{1}{2}} K_{\text{score}}(\phi)^{\frac{1}{2}} \left(\|G^0\|_{W^{1,4}}^4 + 1\right)^{\frac{1}{2}}$$

$$\leq \frac{K_0(\phi)}{n_0}\left(\|G^0\|_{W^{1,4}}^4 + 1\right) \quad (25)$$

where $K_0(\phi)$ is used to combine all of the constants. This now concludes the first part of our results, which bounds the variance of the first term in the MLMC telescoping sum.

We can now derive a similar bound for the other terms (i.e. to simplify (9)). We do this by first only considering the norm inside of the expectation:

$$\left\|\nabla_z f_\phi^l(z) - \nabla_z f_\phi^{l-1}(z)\right\|_2 = \left\|\nabla_z - \log \tilde{q}_\phi\left(g^l(z)\right) - \left(\nabla_z - \log \tilde{q}_\phi\left(g^{l-1}(z)\right)\right)\right\|_2$$

$$= \left\|\nabla_z \log \tilde{q}_\phi\left(g^l(z)\right) - \nabla_z \log \tilde{q}_\phi\left(g^{l-1}(z)\right)\right\|_2$$

$$= \left\|\nabla_z g^l(z)\nabla \log \tilde{q}_\phi(g^l(z)) - \nabla_z g^{l-1}(z)\nabla \log \tilde{q}_\phi\left(g^{l-1}(z)\right)\right\|_2 \quad (26)$$

$$\leq \left\|\nabla_z g^l(z)\left(\nabla \log \tilde{q}_\phi\left(g^l(z)\right) - \nabla \log \tilde{q}_\phi\left(g^{l-1}(z)\right)\right)\right\|_2$$

$$+ \left\|\left(\nabla_z g^l(z) - \nabla_z g^{l-1}(z)\right)\nabla \log \tilde{q}_\phi\left(g^{l-1}(z)\right)\right\|_2 \quad (27)$$

$$\leq \left\|\nabla_z g^l(z)\right\|_2 \left\|\nabla \log \tilde{q}_\phi(g^l(z)) - \nabla \log \tilde{q}_\phi(g^{l-1}(z))\right\|_2$$

$$+ \left\|\nabla_z g^l(z) - \nabla_z g^{l-1}(z)\right\|_2 \left\|\nabla \log \tilde{q}_\phi(g^{l-1}(z))\right\|_2 \quad (28)$$

Here, (26) follows from the chain rule, (27) follows by adding and subtracting the term $\nabla_z g^l(z)\nabla \log \tilde{q}_\phi(g^{l-1}(z))$ and using the triangle inequality, and (28) follows from the Cauchy-Schwarz inequality of the 2-norm. Squaring both sides of this inequality and taking expectations, we

then obtain:

$$
\mathbb{E}\left[\left\|\nabla_z f_\phi^l(z) - \nabla_z f_\phi^{l-1}(z)\right\|_2^2\right] \leq \mathbb{E}\left[\left(\left\|\nabla_z g^l(z)\right\|_2 \left\|\nabla \log \tilde{q}_\phi(g^l(z)) - \nabla \log \tilde{q}_\phi(g^{l-1}(z))\right\|_2\right.\right.
$$
$$
\left.\left. + \left\|\nabla_z g^l(z) - \nabla_z g^{l-1}(z)\right\|_2 \left\|\nabla \log \tilde{q}_\phi\left(g^{l-1}(z)\right)\right\|_2\right)^2\right]
$$
$$
\leq \mathbb{E}\left[2\left\|\nabla_z g^l(z)\right\|_2^2 \left\|\nabla \log \tilde{q}_\phi(g^l(z)) - \nabla \log \tilde{q}_\phi(g^{l-1}(z))\right\|_2^2\right.
$$
$$
\left. + 2\left\|\nabla_z g^l(z) - \nabla_z g^{l-1}(z)\right\|_2^2 \left\|\nabla \log \tilde{q}_\phi\left(g^{l-1}(z)\right)\right\|_2^2\right]. \tag{29}
$$
$$
\leq 2\mathbb{E}\left[\left\|\nabla_z g^l(z)\right\|_2^4\right]^{\frac{1}{2}} \mathbb{E}\left[\left\|\nabla \log \tilde{q}_\phi(g^l(z)) - \nabla \log \tilde{q}_\phi(g^{l-1}(z))\right\|_2^4\right]^{\frac{1}{2}}
$$
$$
+ 2\mathbb{E}\left[\left\|\nabla_z g^l(z) - \nabla_z g^{l-1}(z)\right\|_2^4\right]^{\frac{1}{2}} \mathbb{E}\left[\left\|\nabla \log \tilde{q}_\phi\left(g^{l-1}(z)\right)\right\|_2^4\right]^{\frac{1}{2}}.
$$
$$
\tag{30}
$$

where (29) follows from that fact that for $a, b \in \mathbb{R}$, we have $(a+b)^2 \leq 2a^2 + 2b^2$, and (30) follows from the Cauchy-Schwarz inequality for expectations. To conclude this proof, we notice that the derivation from (13) to (16) can be modified by replacing $G^0$ by $G^l$ gives:

$$
\mathbb{E}\left[\left\|\nabla_z g^l(z)\right\|_2^4\right] \leq K_{\text{grad}}\left(\left\|G^l\right\|_{W^{1,4}}^4 + 1\right) \leq K_{\text{grad}}\left(S_l^4 + 1\right). \tag{31}
$$

where the last inequality holds thanks to Assumption (A2). Similarly, replacing $G^0$ by $G^l - G^{l-1}$ and following the derivations from (13) to (20) gives

$$
\mathbb{E}\left[\left\|\nabla_z g^l(z) - \nabla_z g^{l-1}(z)\right\|_2^4\right] \leq K_{\text{grad}}\left\|G^l - G^{l-1}\right\|_{W^{1,4}}^4. \tag{32}
$$

We notice that we lose the additive term since the last columns of the matrix $\nabla_z g^l(z) - \nabla_z g^{l-1}(z)$ form a zero matrix. This is because

$$
\nabla_z g^l(z) - \nabla_z g^{l-1}(z) \tag{33}
$$
$$
= \begin{bmatrix} \nabla_u G_\theta^l(u_1) - \nabla_u G_\theta^{l-1}(u_1) \cdots \nabla_u G_\theta^l(u_m) - \nabla_u G_\theta^{l-1}(u_m) & \nabla_u \theta - \nabla_u \theta \\ \nabla_\theta G_\theta^l(u_1) - \nabla_\theta G_\theta^{l-1}(u_1) \cdots \nabla_\theta G_\theta^l(u_m) - \nabla_\theta G_\theta^{l-1}(u_m) & \nabla_\theta \theta - \nabla_\theta \theta \end{bmatrix}
$$
$$
= \begin{bmatrix} \nabla_u G_\theta^l(u_1) - \nabla_u G_\theta^{l-1}(u_1) \cdots \nabla_u G_\theta^l(u_m) - \nabla_u G_\theta^{l-1}(u_m) & \mathbf{0} \\ \nabla_\theta G_\theta^l(u_1) - \nabla_\theta G_\theta^{l-1}(u_1) \cdots \nabla_\theta G_\theta^l(u_m) - \nabla_\theta G_\theta^{l-1}(u_m) & \mathbf{0} \end{bmatrix}
$$

The non-zero lower-right block, i.e., $\nabla_\theta \theta$ in (14), was the cause of the additive term in (20) and the final result (the upper-right block is always zero since $\nabla_u \theta = \mathbf{0}$), which is now cancelled out.

Additionally, we could replace $G^0$ by $G^{l-1}$ in the the bound from (22) to (24) in order to get:

$$
\mathbb{E}\left[\left\|\nabla \log q_\phi\left(g^{l-1}(z)\right)\right\|_2^4\right] \leq K_{\text{score}}(\phi)\left(\left\|G^{l-1}\right\|_{W^{1,4}}^4 + 1\right) \tag{34}
$$
$$
\leq K_{\text{score}}(\phi)\left(S_{l-1}^4 + 1\right) \tag{35}
$$

where once again we used Assumption (A2).

Finally, we split the following expression to make use of our local Lipschitz property:

$$\mathbb{E}\left[\left\|\nabla \log \tilde{q}_\phi\left(g^l(z)\right) - \nabla \log \tilde{q}_\phi\left(g^{l-1}(z)\right)\right\|_2^4\right] \tag{36}$$

$$= \mathbb{E}\left[\left\|\nabla \log \tilde{q}_\phi\left(g^l(z)\right) - \nabla \log \tilde{q}_\phi\left(g^{l-1}(z)\right)\right\|_2^4 \,\Big|\, \left\|g^l(z) - g^{l-1}(z)\right\|_2^4 \geq \delta\right]$$
$$\times \mathbb{P}\left[\left\|g^l(z) - g^{l-1}(z)\right\|_2^4 \geq \delta\right]$$
$$+ \mathbb{E}\left[\left\|\nabla \log \tilde{q}_\phi\left(g^l(z)\right) - \nabla \log \tilde{q}_\phi\left(g^{l-1}(z)\right)\right\|_2^4 \,\Big|\, \left\|g^l(z) - g^{l-1}(z)\right\|_2^4 < \delta\right]$$
$$\times \mathbb{P}\left[\left\|g^l(z) - g^{l-1}(z)\right\|_2^4 < \delta\right] \tag{37}$$

$$\leq \frac{8}{\delta}\left(\mathbb{E}\left[\left\|\nabla \log \tilde{q}_\phi\left(g^l(z)\right)\right\|_2^4\right] + \mathbb{E}\left[\left\|\nabla \log \tilde{q}_\phi\left(g^{l-1}(z)\right)\right\|_2^4\right]\right)\mathbb{E}\left[\left\|g^l(z) - g^{l-1}(z)\right\|_2^4\right]$$
$$+ K_{\mathrm{Lip}}(\phi)^4\mathbb{E}\left[\left\|g^l(z) - g^{l-1}(z)\right\|_2^4\right] \times 1 \tag{38}$$

$$\leq \left(\frac{8}{\delta}K_{\mathrm{score}}(\phi)(S_{l-1}^4 + S_l^4 + 2) + K_{\mathrm{Lip}}(\phi)^4\right)\mathbb{E}\left[\left\|g^l(z) - g^{l-1}(z)\right\|_2^4\right] \tag{39}$$

$$\leq K_{\mathrm{score\text{-}diff}}(\phi, \delta)\left\|G^l - G^{l-1}\right\|_{W^{1,4}}^4. \tag{40}$$

Here, (37) follows due to the law of total expectation. For (38), the bound on the first term follows due to $\|a - b\|_2^4 \leq 8(\|a\|_2^4 + \|b\|_2^4)$ and Markov's inequality, and the bound on the second term follows due to the local-Lipschitz condition (i.e. Assumption (A3)) and the fact that a probability is always upper bounded by 1. Then, (39) follows using (35). Finally, (40) follows by grouping all constants together and noting that

$$\mathbb{E}\left[\left\|g^l(z) - g^{l-1}(z)\right\|_2^4\right] = \mathbb{E}\left[\left(\sum_{i=1}^m \left\|G^l(u_i) - G^{l-1}(u_i)\right\|_2^2\right)^2\right] \tag{41}$$

$$\leq m\sum_{i=1}^m \mathbb{E}\left[\left\|G^l(u_i) - G^{l-1}(u_i)\right\|_2^4\right] \leq m\|G^l - G^{l-1}\|_{W^{1,4}}^4 \tag{42}$$

Combining all of the above (i.e. (7), (9), (28), (31), (32), (35), and (39)), we end up with

$$\mathrm{Var}\left[h_l(\phi)\right] \leq \frac{2}{n_l}K_{\mathrm{Poin}}\left(K_{\mathrm{grad}}^{\frac{1}{2}}(S_l^4 + 1)^{\frac{1}{2}}K_{\mathrm{score\text{-}diff}}^{\frac{1}{2}}(\phi, \delta)\left\|G^l - G^{l-1}\right\|_{W^{1,4}}^2\right.$$
$$\left. + K_{\mathrm{grad}}^{\frac{1}{2}}\left(\left\|G^l - G^{l-1}\right\|_{W^{1,4}}^4\right)^{\frac{1}{2}}K_{\mathrm{score}}(\phi)^{\frac{1}{2}}\left(S_{l-1}^4 + 1\right)^{\frac{1}{2}}\right)$$
$$\leq \frac{K_l(\phi)}{n_l}\left\|G^l - G^{l-1}\right\|_{W^{1,4}}^2,$$

where $K_l(\phi)$ combines all constants. This proves our second result and therefore concludes our proof. $\qquad\square$

## A.2   Proof of Theorem 2

*Proof.* We first recall both the cost and the variance of our estimator, modifying our notation slightly to emphasise the number of samples $n_0, \ldots, n_L$. The total cost of this method is given by

$$\mathrm{Cost}(\ell_{\mathrm{MLMC}}(\phi); n_0, \ldots, n_L) = n_0 C_0 + \sum_{l=1}^L n_l(C_l + C_{l-1}),$$

where $C_l$ is the cost of evaluating $f^l$ at level $l$. In addition, we also have the following upper bound on the variance using Theorem 1:

$$\mathrm{Var}\left[\ell_{\mathrm{MLMC}}(\phi); n_0, \ldots, n_L\right] \leq \frac{K_0(\phi)}{n_0}\left(\left\|G^0\right\|_{W^{1,4}}^4 + 1\right) + \sum_{l=1}^L \frac{K_l(\phi)}{n_l}\left\|G^l - G^{l-1}\right\|_{W^{1,4}}^2,$$

where once again we write $\|\cdot\|_{W^{1,4}}$ to denote the norm $\|\cdot\|_{W^{1,4}(\pi \times \mathbb{U})}$. Overall, we would like to solve the following optimisation problem:

$$(n_0^\star, \ldots, n_L^\star)^\top := \arg \min_{(n_0, \ldots, n_L)^\top} \mathrm{Var}\left[\ell_{\mathrm{MLMC}}(\phi); n_0, \ldots, n_L\right]$$

$$\text{such that} \quad \mathrm{Cost}(\ell_{\mathrm{MLMC}}(\phi); n_0, \ldots, n_L) \leq C_{\mathrm{budget}},$$

where $C_{\mathrm{budget}}$ is the computational budget. We relax the problem slightly by minimising the upper bound on the variance instead, thus the problem can be expressed in the following Lagrangian form:

$$
\begin{aligned}
\mathcal{L}(n_0, &\ldots, n_L, \nu) \\
:=& \frac{K_0(\phi)}{n_0}\left(\|G^0\|_{W^{1,4}}^4 + 1\right) + \sum_{l=1}^L \frac{K_l(\phi)}{n_l}\|G^l - G^{l-1}\|_{W^{1,4}}^2 \\
& + \nu\left(\mathrm{Cost}(\ell_{\mathrm{MLMC}}(\phi); n_0, \ldots, n_L) - C_{\mathrm{budget}}\right) \\
=& \frac{K_0(\phi)}{n_0}\left(\|G^0\|_{W^{1,4}}^4 + 1\right) + \sum_{l=1}^L \frac{K_l(\phi)}{n_l}\|G^l - G^{l-1}\|_{W^{1,4}}^2 \\
& + \nu\left(n_0 C_0 + \sum_{l=1}^L n_l(C_l + C_{l-1}) - C_{\mathrm{budget}}\right).
\end{aligned}
$$

This can be solved by setting all partial derivatives with respect to $(n_0, \ldots, n_L)$ and $\nu$ equal to zero and solving the associated system of equations. Firstly, we get:

$$\frac{\partial \mathcal{L}(n_0, \ldots, n_L, \nu)}{\partial n_0} = -K_0(\phi)\left(\|G^0\|_{W^{1,4}}^4 + 1\right) n_0^{-2} + \nu C_0 = 0 \tag{43}$$

$$\Leftrightarrow \quad n_0^\star = \sqrt{\frac{K_0(\phi)}{\nu C_0}\left(\|G^0\|_{W^{1,4}}^4 + 1\right)}, \tag{44}$$

and for $l \in \{1, \ldots, L\}$, we have:

$$\frac{\partial \mathcal{L}(n_0, \ldots, n_L, \nu)}{\partial n_l} = -K_l(\phi)\|G^l - G^{l-1}\|_{W^{1,4}}^2 n_l^{-2} + \nu(C_l + C_{l-1}) = 0$$

$$\Leftrightarrow \quad n_l^\star = \sqrt{\frac{K_l(\phi)}{\nu(C_l + C_{l-1})}\|G^l - G^{l-1}\|_{W^{1,4}}^2}. \tag{45}$$

Finally, taking the partial derivative with respect to $\nu$ confirms that our constraint is active (i.e. we are on the boundary of the feasible region):

$$\frac{\partial \mathcal{L}(n_0, \ldots, n_L, \nu)}{\partial \nu} = n_0 C_0 + \sum_{l=1}^L n_l(C_l + C_{l-1}) - C_{\mathrm{budget}} = 0$$

$$\Leftrightarrow \quad C_{\mathrm{budget}} = n_0 C_0 + \sum_{l=1}^L n_l(C_l + C_{l-1}). \tag{46}$$

Plugging the results of (44) and (45) into (46), we get:

$$C_{\text{budget}} = n_0 C_0 + \sum_{l=1}^{L} n_l (C_l + C_{l-1})$$

$$= \sqrt{\frac{K_0(\phi)}{\nu C_0} \left( \|G^0\|_{W^{1,4}}^4 + 1 \right)} \times C_0$$

$$+ \sum_{l=1}^{L} \sqrt{\frac{K_l(\phi)}{\nu(C_l + C_{l-1})} \|G^l - G^{l-1}\|_{W^{1,4}}^2} \times (C_l + C_{l-1})$$

$$= \nu^{-\frac{1}{2}} \left( \sqrt{K_0(\phi) C_0 \left( \|G^0\|_{W^{1,4}}^4 + 1 \right)} \right.$$

$$\left. + \sum_{l=1}^{L} \sqrt{K_l(\phi)(C_l + C_{l-1}) \|G^l - G^{l-1}\|_{W^{1,4}}^2} \right),$$

which gives

$$\nu = \frac{\left( \sqrt{K_0(\phi) C_0 \left( \|G^0\|_{W^{1,4}}^4 + 1 \right)} + \sum_{l=1}^{L} \sqrt{K_l(\phi)(C_l + C_{l-1}) \|G^l - G^{l-1}\|_{W^{1,4}}^2} \right)^2}{C_{\text{budget}}^2}. \quad (47)$$

We can now use the expression for $\nu$ that we obtained in (47) to obtain a simplified expression for $n_0, \ldots, n_L$ (using (44) and (45)):

$$n_0^\star \propto \frac{C_{\text{budget}}}{\sqrt{C_0}} \sqrt{\|G^0\|_{W^{1,4}}^4 + 1}, \qquad n_l^\star \propto \frac{C_{\text{budget}}}{\sqrt{C_l + C_{l-1}}} \|G^l - G^{l-1}\|_{W^{1,4}} \quad \text{for } l \in \{1, \ldots, L\}.$$

This completes our proof. $\qquad \square$

### A.3 Extension of Theorem 1 to the gradient

We provide an upper bound on the variance for each element of the gradient $\nabla_\phi \ell_{\text{MLMC}}(\phi)$, that is, on each partial derivative $\nabla_{\phi_j} \ell_{\text{MLMC}}(\phi)$ for $j \in \{1, \ldots, d_\phi\}$. The partial derivatives are given by

$$\nabla_{\phi_j} \ell_{\text{MLMC}}(\phi) = \nabla_{\phi_j} h_0(\phi) + \sum_{l=1}^{L} \nabla_{\phi_j} h_l(\phi)$$

$$= \frac{1}{n_0} \sum_{i=1}^{n_0} \nabla_{\phi_j} f_\phi^0(u_i^0, \theta_i^0) + \sum_{l=1}^{L} \frac{1}{n_l} \sum_{i=1}^{n_l} (\nabla_{\phi_j} f_\phi^l(u_i^l, \theta_i^l) - \nabla_{\phi_j} f^{l-1}(u_i^l, \theta_i^l))$$

**Theorem 3.** *Let $\phi \in \Phi \subseteq \mathbb{R}^{d_\Phi}$ and suppose the following assumption hold in addition to A1-A2 in theorem 1:*

(A3') $\nabla_{\phi_j} \log \tilde{q}_\phi$ *is continuously differentiable, locally $K_{Lip}^j(\phi)-$smooth and satisfies the growth condition $\|\nabla_{\phi_j} \nabla \log \tilde{q}_\phi(x_{1:m}, \theta)\|_2 \leq K_{grow}^j(\phi)(\sum_{i=1}^{m} \|x_i\|_2 + \|\theta\|_2 + 1)$ for some $K_{Lip}^j(\phi), K_{grow}^j(\phi) > 0$ for all $j = 1, \ldots, d_\Phi$.*

*Then, for $l \in \{1, \ldots, L\}$, $K_0^j(\phi), \ldots, K_L^j(\phi) > 0$, $j \in \{1, \ldots, d_\phi\}$ independent of $n_0, \ldots, n_L$, we have that:*

$$Var\left[ \nabla_{\phi_j} h_0(\phi) \right] \leq \frac{K_0^j(\phi)}{n_0} \left( \|G^0\|_{W^{1,4}(\pi \times \mathbb{U})}^4 + 1 \right),$$

$$\text{and } Var\left[ \nabla_{\phi_j} h_l(\phi) \right] \leq \frac{K_l^j(\phi)}{n_l} \left( \|G^l - G^{l-1}\|_{W^{1,4}(\pi \times \mathbb{U})}^2 \right)$$

*Proof.* The variance of each term can be expressed as

$$\text{Var}\left[\nabla_{\phi_j} h_0(\phi)\right] = \frac{1}{n_0}\text{Var}[\nabla_{\phi_j} f_\phi^0(z)] \tag{48}$$

$$\text{Var}\left[\nabla_{\phi_j} h_l(\phi)\right] = \frac{1}{n_l}\text{Var}[\nabla_{\phi_j} f_\phi^l(z) - \nabla_{\phi_j} f_\phi^{l-1}(z)] \tag{49}$$

Assume that $\nabla_{\phi_j} f_\phi$ is sufficiently regular, applying Poincaré inequality to (48) and (49) (as in (8) and (9)) gives:

$$\text{Var}[\nabla_{\phi_j} f_\phi^0(z)] \le K_{\text{Poin}}\mathbb{E}\left[\left\|\nabla_z\nabla_{\phi_j} f_\phi^0(z)\right\|_2^2\right] \tag{50}$$

$$\text{Var}[\nabla_{\phi_j} f_\phi^l(z) - \nabla_{\phi_j} f_\phi^{l-1}(z)] \le K_{\text{Poin}}\mathbb{E}\left[\left\|\nabla_z\nabla_{\phi_j} f_\phi^l(z) - \nabla_z\nabla_{\phi_j} f_\phi^{l-1}(z)\right\|_2^2\right] \tag{51}$$

where the expectation is over $z$. We can simplify (50) as

$$\mathbb{E}\left[\left\|\nabla_z\nabla_{\phi_j} f_\phi^0(z)\right\|_2^2\right] = \mathbb{E}\left[\left\|\nabla_z\nabla_{\phi_j} \log\tilde{q}_\phi(g^0(z))\right\|_2^2\right] \tag{52}$$

$$= \mathbb{E}\left[\left\|\nabla_z g^0(z)\nabla_{\phi_j} \log\tilde{q}_\phi(g^0(z))\right\|_2^2\right] \tag{53}$$

$$\le \mathbb{E}\left[\left\|\nabla_z g^0(z)\right\|_2^4\right]^{\frac{1}{2}} \mathbb{E}\left[\left\|\nabla\nabla_{\phi_j} \log\tilde{q}_\phi(g^0(z))\right\|_2^4\right]^{\frac{1}{2}} \tag{54}$$

Here (53) is due to the chain rule and (54) follows (11) - (12). The bound for the first expectation is given by (20). For the second expectation, we have:

$$\mathbb{E}\left[\left\|\nabla\nabla_{\phi_j} \log\tilde{q}_\phi(g^0(z))\right\|_2^4\right] \le K_{\text{score}}^j(\phi)\left(\left\|G^0\right\|_{W^{1,4}}^4 + 1\right) \tag{55}$$

by Assumption (A3') and following (21) - (24). Combining (48), (50), (54), (20), and (55) gives:

$$\text{Var}[\nabla_{\phi_j} h_0(\phi)] \le \frac{K_0^j(\phi)}{n_0}\left(\left\|G^0\right\|_{W^{1,4}}^4 + 1\right) \tag{56}$$

where $K_0^j(\phi)$ is a constant depending on $\phi$ and $j$, independent of $n_0$. Now, we derive an upper bound on $\text{Var}\left[\nabla_{\phi_j} h_l(\phi)\right]$. Following (26) - (30), we have:

$$\mathbb{E}\left[\left\|\nabla_z\nabla_{\phi_j} f_\phi^l(z) - \nabla_z\nabla_{\phi_j} f_\phi^{l-1}(z)\right\|_2^2\right] \tag{57}$$

$$\le 2\left[\left(\mathbb{E}\left[\left\|\nabla_z g^l(z)\right\|_4^2\right]\right)^{\frac{1}{2}}\left(\mathbb{E}\left[\left\|\nabla\nabla_{\phi_j} \log\tilde{q}_\phi(g^l(z)) - \nabla\nabla_{\phi_j} \log\tilde{q}_\phi(g^{l-1}(z))\right\|_2^4\right]\right)^{\frac{1}{2}}\right.$$

$$\left. + \mathbb{E}\left[\left\|\nabla_z g^l(z) - \nabla_z g^{l-1}(z)\right\|_2^4\right]^{\frac{1}{2}}\mathbb{E}\left[\left\|\nabla\nabla_{\phi_j} \log\tilde{q}_\phi(g^{l-1}(z))\right\|_2^4\right]^{\frac{1}{2}}\right] \tag{58}$$

Bound for the first expectation is given by (31) and the third expectation is given by (32). For the forth expectation, from (55) (replacing 0 with $l$) and Assumption (A2), we have:

$$\mathbb{E}\left[\left\|\nabla\nabla_{\phi_j} \log\tilde{q}_\phi\left(g^{l-1}(z)\right)\right\|_2^4\right] \le K_{\text{score}}^j(\phi)\left(S_{l-1}^4 + 1\right) \tag{59}$$

For the second expectation, following (36)-(40), and using our local Lipschitz property, we get:

$$\mathbb{E}\left[\left\|\nabla\nabla_{\phi_j}\log\tilde{q}_\phi(g^l(z)) - \nabla\nabla_{\phi_j}\log\tilde{q}_\phi(g^{l-1}(z))\right\|_2^4\right] \tag{60}$$

$$\leq K_{\text{score-diff}}^j(\phi,\delta)\left\|G^l - G^{l-1}\right\|_{W^{1,4}}^4 \tag{61}$$

where $K_{\text{score-diff}}^j(\phi)$ combines all the constant, independent of $n_l$ and the difference in the simulators.

Finally, applying the bounds for the each expectation; (31), (61), (32), and (59), to (58), and combining it with (49) gives:

$$\text{Var}[\nabla_{\phi_j}h_0(\phi)] \leq \frac{2K_{\text{Poin}}}{n_l}\Big[K_{\text{grad}}^{\frac{1}{2}}(S_l^4 + 1)^{\frac{1}{2}}(K_{\text{score-diff}}^j(\phi))^{\frac{1}{2}}\|G^l - G^{l-1}\|_{W^{1,4}}^2$$
$$+ K_{\text{grad}}^{\frac{1}{2}}\left(\|G^l - G^{l-1}\|_{W^{1,4}}^4\right)^{\frac{1}{2}}(K_{\text{score}}^j(\phi))^{\frac{1}{2}}(S_{l-1}^4 + 1)^{\frac{1}{2}}\Big] \tag{62}$$

$$\leq \frac{K_l^j(\phi)}{n_l}\left(\|G^l - G^{l-1}\|_{W^{1,4}}^2\right) \tag{63}$$

where $K_l$ combines all the constant. Combining (56) and (63) gives a bound on the variance of the partial derivative:

$$\text{Var}[\nabla_{\phi_j}\ell_{\text{MLMC}}(\phi)] \leq \frac{K_0^j(\phi)}{n_0}\left(\|G^0\|_{W^{1,4}(\pi\times\mathbb{U})}^4 + 1\right) + \sum_{l=1}^L \frac{K_l^j(\phi)}{n_l}\|G^l - G^{l-1}\|_{W^{1,4}(\pi\times\mathbb{U})}^2 \tag{64}$$

$\square$

## B  Experimental details & additional results

For all the experiments, we use a Mac M4 CPU with 16 GB memory for the training of neural networks. Running all the experiments roughly takes half a day. For optimisation, we use batch gradient descent with the Adam optimiser [Kingma and Ba, 2015] using Pytorch [Paszke et al., 2017]. For the construction of the conditional density estimator, we use the SBI package [Tejero-Cantero et al., 2020]. We also use the BayesFlow package [Radev et al., 2023b] for some of the figures. For each experiment, we fix the number of epochs unless stated otherwise. We use the same setting and stopping criterion for the NPE and the NLE baseline as in the default implementation of the SBI package. We set the learning rate to $10^{-4}$ for all the experiments.

### B.1  The g-and-k distribution

**Simulator setup.** The g-and-k distribution can be written in simulator form as follows:

$$G_\theta(u) = \theta_1 + \theta_2\left(1 + 0.8\left(\frac{1 - \exp(-\theta_3 z(u))}{1 + \exp(-\theta_3 z(u))}\right)\right)\left(1 + z(u)^2\right)^{\log(\theta_4)}z(u),$$
$$z(u) = \Phi^{-1}(u) = \sqrt{2}\text{erf}^{-1}(2u - 1), \qquad u \sim \text{Unif}([0,1]),$$

where $\Phi^{-1}$ is the quantile function of the standard normal distribution and $\text{erf}^{-1}$ is the inverse of error function. We set the prior distribution to be a tensor product of marginal distributions on each parameter: $\theta_1, \theta_2, \theta_3 \sim \text{Unif}([0,3]^3)$ and $\theta_4 \sim \text{Unif}([0,\exp(0.5)])$. Note that we always resort to an approximation method for the evaluation of $\text{erf}^{-1}(\cdot)$. For the high-fidelity simulator, we use an accurate approximation implemented in Scipy [Virtanen et al., 2020]. For the low-fidelity simulator, we use a Taylor expansion of $\text{erf}^{-1}$ up to the third order:

$$z_{\text{low}}(u) := \sqrt{2}\text{erf}_{\text{low}}^{-1}(2u - 1), \quad \text{erf}_{\text{low}}^{-1}(v) := \frac{\pi}{2}\left(u + \frac{\pi}{12}u^3\right).$$

An advantage of the g-and-k distribution is that its density can be approximated numerically almost exactly. Following Rayner and MacGillivray [2002], we first numerically obtain $F_\theta(x) = G_\theta^{-1}(x)$

by solving numerically for $x_i - G_\theta(u_i) = 0$ using a root-solving algorithm [Press, 2007][1], then we obtain the density function by taking $\hat{p}(x \mid \theta) := F'_\theta(x) = \partial F_\theta(x)/\partial x$. For this second step, we typically use a finite difference approximation. Seed-matching is simply done by using same the $u$ and $\theta$.

**Neural network details**

**NLE:** We use the neural spline flow (NSF) [Durkan et al., 2019]. We pick 10 bins, span of $[-7, 7]$ and 1 coupling layer since we only have one dimensional input. The conditioner for the NSF is a multilayer perceptron neural network (MLP) with 3 hidden layers of 50 units, and 10% dropout, trained for 10,000 epochs.

**NPE:** We use NSF with 3 bins, span of $[-3, 3]$ and 3 coupling layers. The conditioner for the NSF is a MLP with 2 hidden layers of 50 units, and 10% dropout, trained for 800 epochs. For each true parameter values, we produce $m = 1000$ iid samples and obtain quantile based 4-dimensional summary statistics following Prangle [2016].

**Evaluations details**

**NLE:** We calculate the forward KL-divergence between the almost exact density and the approximated density over 2000 equidistant points in $[-30, 30]$.

**NPE:** We calculate NLPD over 500 simulations. Empirical coverage is estimated using average value of 500 simulated datasets, where we draw 2000 posterior samples from each simulations. We then calculate $1 - \beta$-highest posterior density credible interval, where $\beta$ is 101 equidistant points between 0 and 1.

## B.2 Training time comparison

We report training time per epoch for our multilevel versions of NLE and NPE and their standard counterparts for the g-and-k experiment; see Appendix B.2. For both methods, we picked $n = 2000$ for the standard NLE/NPE with MC loss and $n_0 = 1000, n_1 = 500$ for our ML-NLE/ML-NPE with MLMC loss such that the total number of samples are the same. Each network is trained independently 10 times to assess uncertainty, with a training budget of 500 epochs for NLE and 100 epochs for NPE.

| Method | Training Time per Epoch (s $\times 10^{-3}$) |
|--------|--------------------------------------------|
| ML-NLE | 1.82 (±0.05) |
| NLE | 1.58 (±0.06) |
| ML-NPE | 8.04 (±0.17) |
| NPE | 6.56 (±0.14) |

Table 1: Average training time per epoch (standard deviation in gray).

## B.3 Ornstein–Uhlenbeck process

**Simulator setup.** Given $T = 10$ (total time), $\Delta t = 0.1$ (time step), $x_0 = 2.0$ (initial value), $\theta = [\gamma, \mu, \sigma]^\top$, $N = \lfloor T/\Delta t \rfloor = 100$ (number of steps), the high- and the low-fidelity OUP simulators are defined as follows:

**High-fidelity:**

For $t = 0, \ldots N - 1$

$$x_{t+1} = x_t + \Delta x_t, \quad \Delta x_t = \gamma(\mu - x_t) + \sigma u_t \sqrt{\Delta t}, \quad u_t \sim \mathcal{N}(0, 1)$$

---

[1]Note that since $G_\theta(u)$ is a quantile function, $F_\theta(x) := G_\theta^{-1}(x)$ is a cumulative distribution function.

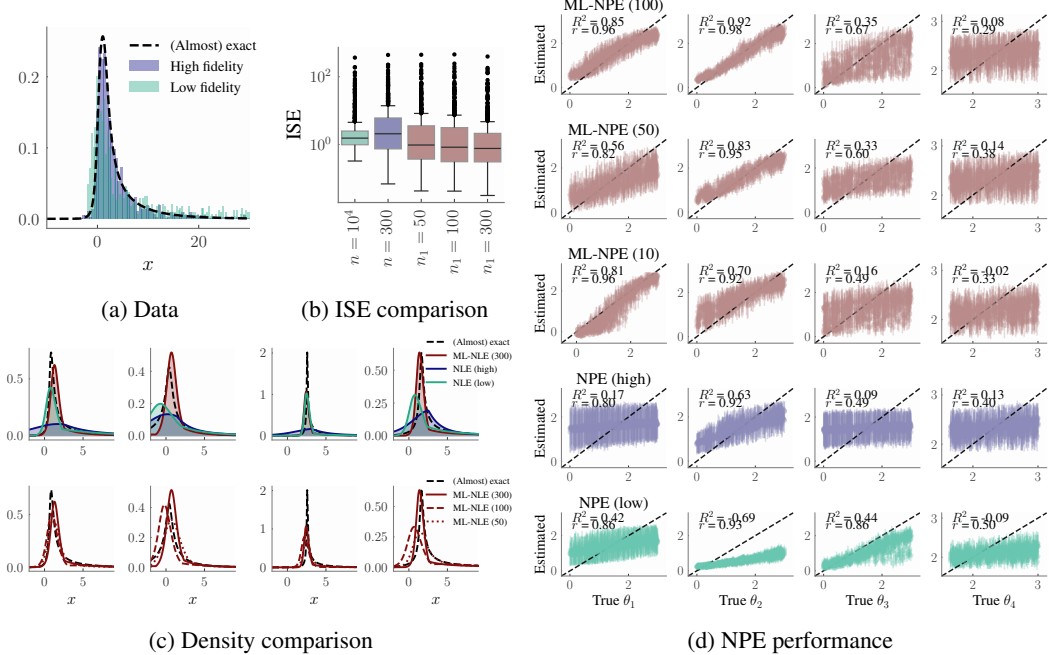

(a) Data

(b) ISE comparison

(c) Density comparison

(d) NPE performance

Figure 6: Additional results for the g-and-k experiment. (a) Histogram of 1000 seed-matched samples from low- and high-fidelity simulator. (b) ISE (integrated squared error) for NLE ($\downarrow$): sum of the squared distance between (almost) exact density and approximated density over 2000 points in the interval $[-30, 30]$. The performance of ML-NLE improves as $n_1$ increases. (c) Approximated density and (almost) exact density for NLE across four simulations. The first row of (c) shows that we get better approximation of the density when combining low and fidelity samples than using them separately. The second row of (c) shows the improvement of the performance of our ML-NLE as $n_1$ increases. These results demonstrate the effectiveness of our method. (d) Recovery plot for NPE: It measures how well the ground truth value is captured by the median of approximated posterior. The x-axis shows the ground truth parameter value and the y-axis shows the median of the posterior distribution with the median absolute deviation, i.e. $\pm\text{median}(|\theta - \text{median}(\theta)|)$, shown around the points. Here $r$ and $R^2$ denotes the correlation coefficient ($\uparrow$) and the coefficient of determination ($\uparrow$) between the ground truth values and the estimated median, respectively. The dashed diagonal line indicates perfect recovery of the true parameter. Using MLMC leads to significant improvements in the parameter recovery, which become more pronounced as $n_1$ increases.

**Low-fidelity:**

$$x_{1:N} = u_{1:N}(\sigma/\sqrt{2\gamma}) + \mu, \quad u_{1:N} \sim \mathcal{N}(0, 1)$$

We set the prior distribution $\theta \sim \text{Unif}([0.1, 1.0] \times [0.1, 3.0] \times [0.1, 0.6])$. The resulting $x$ is a 100-dimensional time series. Seed-matching is simply done by using the same $u$ and $\theta$.

**Neural network details.** We use NSF with 2 bins, span of $[-2, 2]$ and 2 coupling layers. The conditioner for the NSF is a MLP with 1 hidden layer of 32 units, and $10\%$ dropout. For ML-NPE, we trained the network for 500 epochs. For TL-NPE, we set a high maximum number of epochs and used on early stopping. The stopping criterion is based on the validation loss computed on $20\%$ of the data, with a patience parameter controlling how many epochs to wait before stopping. For each true parameter values, we produce $m = 1$ sample and use 5 representative data points as a summary statistics. The subsamples are taken at equal interval in log space such that we take $0, 3, 10, 31, 99$th data points.

**Evaluations details.** We train each ML-NPE and TL-NPE 20 times. For each trained network, we compute the NLPD and KL divergence against reference posterior across 500 simulated datasets.

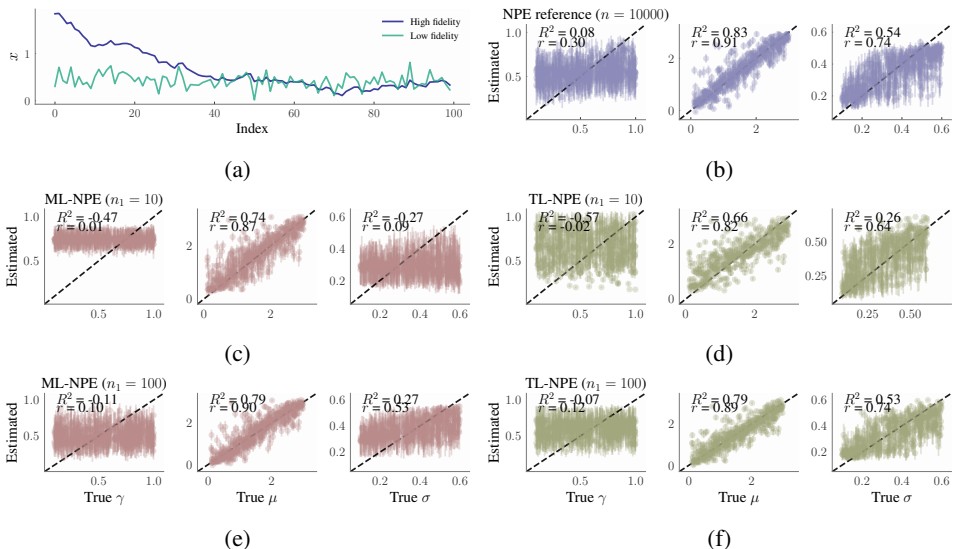

Figure 7: Additional results for the OUP experiment. (a) Example of one seed-matched sample from high and low-fidelity simulators. (b) Recovery plot for the reference NPE posterior with $n = 10^4$. (c)-(f) Recovery plots for ML-NPE and TL-NPE for $n_1 = 10$ and $n_1 = 100$.

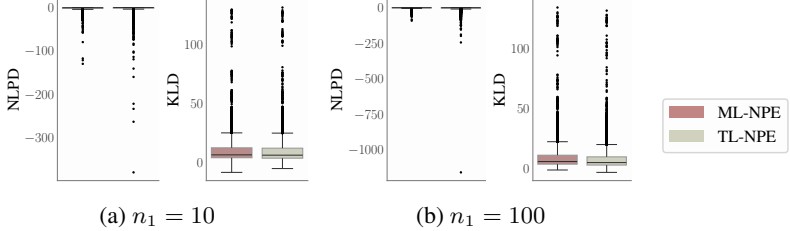

Figure 8: Figure 8 with all the data, without removing outliers. Outliers are identified using IQR method with factor 4, which removes only extreme outliers.

### B.3.1 Experiment with different dimension of $\theta$

We now conduct an experiment to explore a failure mode of our method. We include an additional parameter in the high-fidelity simulator, termed "initial value" $\theta_4 = x_0 \sim \text{uniform}([0, 4])$, instead of fixing it at $x_0 = 2.0$ as we did previously. For this experiment, we picked $n_0 = 1,000$ and $n_1 = 100$. We additionally trained the conditional density estimator using only $n = 100$ data from the high-fidelity simulator. All the other settings remain the same.

Table 2: Mean and standard deviation of NLPD and KLD.

|  | ML-NPE | NPE (high only) |
|---|---|---|
| NLPD $\downarrow$ | -0.18 | **-1.21** |
|  | (0.04) | (0.53) |

We observe that it notably underperforms in NLPD compared to using only high-fidelity data. This may be due to our training objective being more unstable and therefore more sensitive to large differences in simulator outputs introduced by the additional parameter $x_0$.

### B.4 Toggle switch model

**Simulator setup.** We follow the implementation by Key et al. [2025]. Given parameters $\theta = [\alpha_1, \alpha_2, \beta_1, \beta_2, \mu, \sigma, \gamma]^\top$, we can sample from the simulator by $x \sim \widetilde{\mathcal{N}}(\mu + u_T, \mu\sigma/u_T^\gamma)$, where $\widetilde{\mathcal{N}}$ denotes the truncated normal distribution on $\mathbb{R}_+$. Here, $u_T$ is given by the discretised-time equation

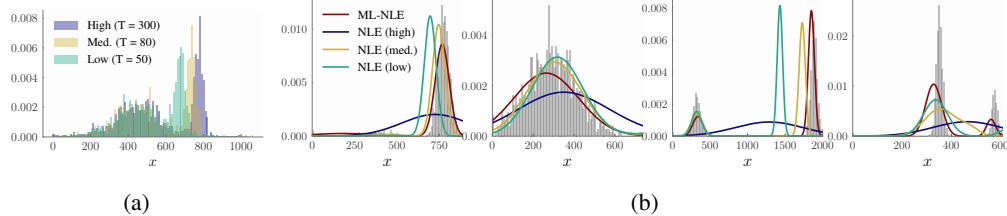

(a)                                                    (b)

Figure 9: Additional results for the Toggle-switch experiment. (a) Histograms of 1000 seed-matched samples from the high-, the medium-, and the low-fidelity simulators. Note that the difference between the low- and the medium-fidelity simulator is larger than the difference between the medium- and the high-fidelity simulator. This suggests that we should take $n_1 > n_2$. (b) Approximated density and samples from high-fidelity simulator across four simulations. ML-NLE demonstrates superior performance compared to the NLE baselines trained exclusively on high-, medium-, and low-fidelity data with the same total simulation budget. In particular, the density estimated by ML-NLE aligns more closely with high-fidelity simulations, highlighting the improved accuracy of our method.

as follows:

For $t = 0, \ldots T - 1$:

$$u_{t+1} \sim \widetilde{\mathcal{N}}(\mu_{u,t}, 0.5), \quad \mu_{u,t} = u_t + \frac{\alpha_1}{\left(1 + v_t^{\beta_1}\right)} - (1 + 0.03 u_t),$$

$$v_{t+1} \sim \widetilde{\mathcal{N}}(\mu_{v,t}, 0.5), \quad \mu_{v,t} = v_t + \frac{\alpha_2}{\left(1 + u_t^{\beta_2}\right)} - (1 + 0.03 v_t)$$

We set the initial state $u_0 = u_1 = 10$ and prior distribution $\theta \sim \text{Unif}([0.01, 50] \times [0.01, 50] \times [0.01, 5] \times [0.01, 5] \times [250, 450] \times [0.01, 0.5] \times [0.01, 0.4])$. The cost of the simulator increases linearly with $T$, and $T \to \infty$ leads to an exact simulation. Different choice of $T$ leads to the simulator with different fidelity levels, having different cost and precision. The expression for the cost of simulating data for the MLMC and MC is given by

$$C_{\text{ML-NLE}} = c \left( n_0 T_0 + \sum_{l=1}^{L} n_l (T_l + T_{l-1}) \right)$$

$$C_{\text{NLE}} = cTn.$$

This yields a total data generation cost for ML-NLE with $(n_0, n_1, n_2) = (10^4, 500, 100)$, and $(T_0, T_1, T_2) = (50, 80, 300)$ as $C_{\text{MLMC}} = 603000$, assuming a unit cost $c = 1$. We then allocate the same computational budget to NLE at different fidelities. This results in the following sample sizes: $n_2 = C_{\text{ML-NLE}}/T_2 = 2010$, $n_1 = C_{\text{ML-NLE}}/T_1 = 7537$, and $n_0 = C_{\text{ML-NLE}}/T_0 = 12060$.

Note that $\mathcal{U}$ varies with the fidelity level of the simulator, as it is given by $\mathbb{R}^{2T+1}$. This discrepancy can complicate seed-matching across fidelities. To address this, we define a unified domain for the noise $\mathcal{U} = \bigcup_{l=0}^{L} \mathcal{U}^l$ shared across all fidelity levels, and assume that for lower fidelities (smaller $l$), certain components of the input are simply disregarded. For seed-matching, we share $\theta$ and $u_l \cap u_{l-1} = u_{l-1}$ where $u_l \in \mathcal{U}^l \subseteq \mathcal{U}, \forall l$.

**Neural network details.** We use Gaussian mixture density network [Bishop, 1994] with two mixture components. We use MLP with 2 hidden layers and 20 units to estimate the parameters: means, variances, and mixture weights, trained for $10,000$ epochs.

**Evaluations details.** We sample $m = 500$ from both high-fidelity simulator and each approximated densities, and calculate MMD over 5000 simulations. The length scale is estimated by the median heuristic.

**B.4.1  Experiment with different allocation of samples** $n_l$

We now repeat the same experiment, but with different allocations of $n_0, n_1, n_2$, keeping the total computational cost roughly the same. Apart from the choice of $n_0 = 10,000, n_1 = 500, n_2 = 100$ that we use in Section 5.3 (option A) which is guided by our theory, we include two other alternatives: (B) $n_0 = 9260, n_1 = 200, n_2 = 300$ and (C) $n_0, n_1, n_2 = 1077$. Option B places more budget on learning the difference between the medium- and the high-fidelity simulator (opposite of option A), and option C allocates equal number of samples for all the terms. The results in Table 3 indicate that option A is the best performing, indicating performance can be improved by careful assignment of the computational budget led by insights from our theory.

Table 3: Mean and standard deviation of MMD. The results for option A, only high, medium, and low remains the same as in Section 5.3.

|  | Option A | Option B | Option C | only high | only medium | only low |
|---|---|---|---|---|---|---|
| MMD ($\downarrow$) | **0.16** | 0.33 | 0.29 | 0.43 | 0.37 | 0.59 |
|  | (0.23) | (0.26) | (0.28) | (0.29) | (0.45) | (0.65) |

## B.5  Cosmological simulations

**Simulator setup.**   We use the CAMELS simulation suite [Villaescusa-Navarro et al., 2021, 2023], a benchmark dataset for machine learning in astrophysics, to study multi-fidelity simulation-based inference in cosmology. The simulation is one of the most expensive cosmological suites ever run; a small fraction of the next generation are being run on the UK DIRAC HPC Facility with 15M CPUh.

The dataset includes both low-fidelity (gravity-only N-body) and high-fidelity (hydrodynamic) simulations of 25 Mpc/$h$ cosmological volumes. The original inference task involves two cosmological parameters, $\theta = (\Omega_m, \sigma_8)$, where $\Omega_m$ is the matter density and $\sigma_8$ the amplitude of fluctuations, with mock power spectrum measurements $P(k)$ used to infer the parameters. However, due to the limited availability of both high- and low-fidelity simulations, we focus on inferring only $\sigma_8$ and treat $\Omega_m$ as part of the nuisance parameter. We found that attempting to jointly infer both parameters led to poor performance (expected due to physical $\sigma_8$-$\Omega_m$ degeneracy) even when using 90% of the high-fidelity data.

In this setup, low-fidelity simulations are governed solely by $\theta$ and the initial condition seed $u$, while high-fidelity simulations additionally incorporate complex astrophysical processes (e.g., feedback), modelled via four extra parameters. These high-fidelity simulations are significantly more computationally expensive—often more than 100× slower to generate than low-fidelity ones. Given these limitations, cosmological analyses often rely on conservative data cuts to exclude small-scale modes (e.g., $k \gtrsim 0.1, h$/Mpc), where simulation inaccuracies are most pronounced [e.g. Jeffrey et al., 2025, Gatti et al., 2025]. Multi-fidelity approaches aim to mitigate such constraints by leveraging both inexpensive and high-fidelity simulations to improve the accuracy of cosmological inference.

Note that the idea of combining low- and high- fidelity simulations has previously been proposed in cosmology; see for example the work of Chartier et al. [2021], Chartier and Wandelt [2022] who use low-fidelity simulations to construct approximations of quantities of interest which can be used as control variates. This work differs from our proposed approach in that it targets quantities such as means and covariances, rather than f the training objective of neural SBI.

**Neural network details.**   We use NSF with 3 bins, span of $[-3, 3]$ and 3 coupling layers. The conditioner for the NSF is an MLP with 2 hidden layer of 30 units, and $10\%$ dropout, trained for 400 epochs.

**Evaluations details.**   We use the same setting as the g-and-k experiment for NPE except that we use 980 test simulations for evaluation.

## B.6  Ablation study of the gradient adjustment technique

We conduct experiments to evaluate the effect of the different gradient adjustment techniques we employ on the performance of our method. We train both ML-NPE and ML-NLE using (i) our gradient adjustment approach which involves both rescaling and projection, (ii) only rescaling, (iii)

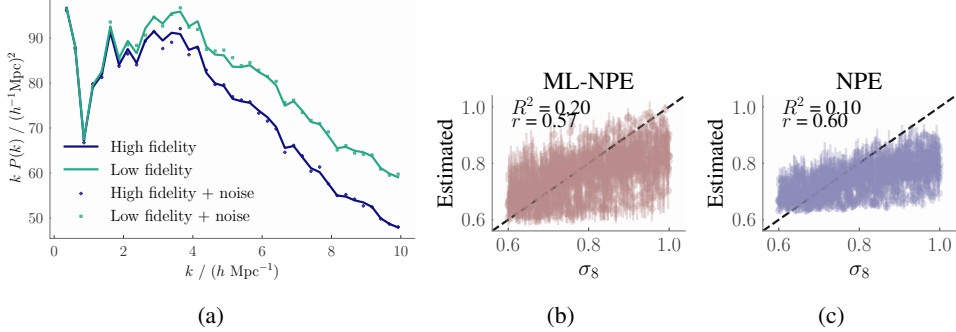

(a)                       (b)                       (c)

Figure 10: Additional results for the cosmological simulator experiment. (a) 1 seed-matched sample from high and low-fidelity simulators. The one with noise is used to train the neural networks. (b)-(c) Recovery plot of ML-NPE and NPE. Adding low-fidelity samples lead better recovery of the parameter.

only projection, and (iv) no gradient adjustment (standard training). We then compare the performance of NPE and NLE on the g-and-k and NLE on the toggle switch experiment. Other than the choice of the gradient adjustment, the training method, hyperparameters, and evaluation methods remain the same. The results are shown in Table 4.

Table 4: Mean and standard deviation of the metrics. For g-and-k NPE, $n_0 = 1000, n_1 = 100, m = 1000$ and for g-and-k NLE, $n_0 = 10^4, n_1 = 300, m = 1$ as in Section 5.1. For toggle switch NLE, $n_0 = 10000, n_1 = 500, n_2 = 100$ as in Section 5.3.

| | (i) both | (ii) only rescaling | (iii) only projection | (iv) standard |
|---|---|---|---|---|
| g-and-k NPE (NLPD ↓) | **-0.30** | -0.21 | -0.11 | -0.13 |
| | (0.31) | (0.25) | (0.11) | (0.27) |
| g-and-k NLE (KLD ↓) | 0.22 | **0.21** | 0.24 | 0.24 |
| | (0.47) | (0.45) | (0.46) | (0.28) |
| Toggle-switch NLE (MMD ↓) | **0.26** | 0.50 | 0.35 | 0.59 |
| | (0.25) | (0.37) | (0.27) | (0.31) |

We observe that when the MLMC loss diverges under standard training, as is the case for the toggle switch experiment and g-and-k with NPE, our gradient adjustment approach that combines both gradient rescaling and projection yields the best results. In the case of NLE on the g-and-k simulator, the loss does not diverge and standard training is sufficient. In such cases, the gradient adjustment yields similar results as standard training, albeit with an increase in the variance of the metric. Therefore, we suggest optimising using our gradient adjustment approach when using ML-NLE or ML-NPE as it avoids the need to first detect whether the loss is diverging or not.

We further compare the training losses with and without gradient adjustment. With the gradient adjustment, the loss diverges, whereas with it, the loss curve exhibits convergence; see Figure 11. To clarify the reason for this behaviour, we also provide an illustration of our gradient scaling procedure in Figure 12.

We note that the optimisation requires some form of regularisation, and the one we used is one possible choice. In our experiments, the gradient adjustment approach performed the best among the options we tried (e.g., regularising the contribution of the difference term when it dominates the first term, or penalising high variance in the difference term).

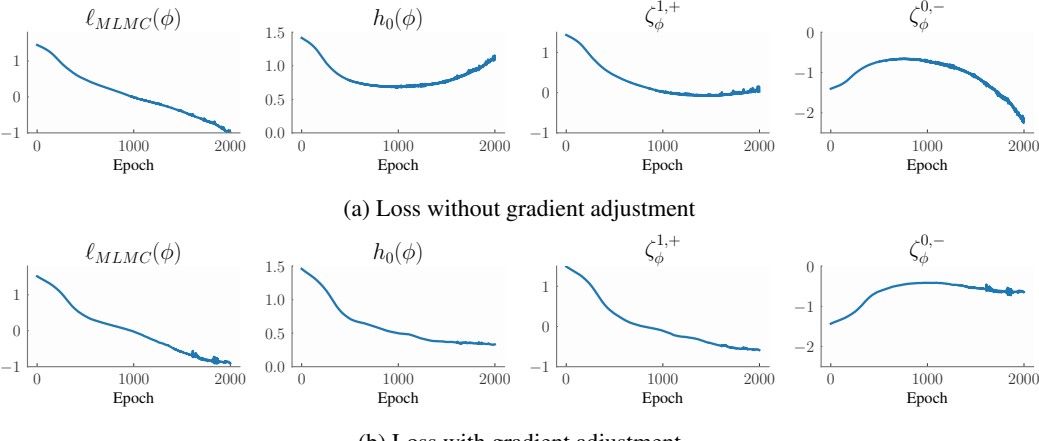

(a) Loss without gradient adjustment

(b) Loss with gradient adjustment

Figure 11: Comparison of training losses with and without gradient adjustment. (a) Without gradient adjustment, the contribution of $\zeta_\phi^{0,-}$ begins to dominate the loss around epoch 1000. The resulting conflict between gradient components leads to unstable optimisation, as evidenced by strong fluctuations in the loss and eventual divergence. (b) With gradient adjustment, all components contribute more stably, and the overall loss decreases steadily eventually, indicating convergence. Loss functions are shown for the g-and-k experiment with NLE.

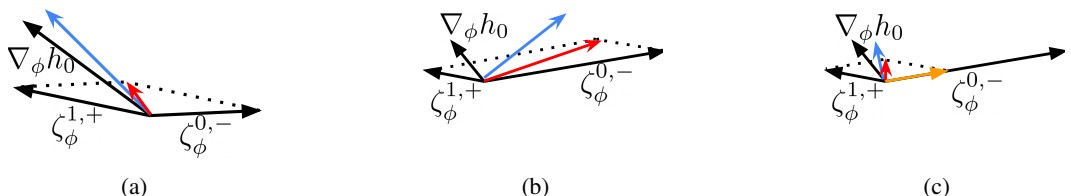

Figure 12: Illustration of the gradient scaling with two levels. The coloured arrows indicate gradient of the correction term, total gradient, and scaled gradient respectively. (a) Ideal case: $||\nabla_\phi h_c(\phi)||$ remains small and works as a correction to $\nabla_\phi h_0(\phi)$. (b) In the later stage of training, $\nabla_\phi h_0(\phi)$ and $\zeta_\phi^{1,+}$ diminishes and $\zeta_\phi^{0,-}$ starts to dominate the optimisation. (c) Gradient scale adjustment: We scale $\zeta_\phi^{0,-}$ such that $||\zeta_\phi^{1,+}|| \approx ||\zeta_\phi^{0,-}||$ and $||\nabla_\phi h_c(\phi)||$ remains small throughout the training as intended.

