# OpenReview forum: "Multilevel neural simulation-based inference"
_NeurIPS.cc/2025/Conference — NeurIPS 2025 poster_

### Official Review · Reviewer_PW7W · 2025-06-02

**Clarity:** 4
**Significance:** 4
**Originality:** 2
**Rating:** 5
**Confidence:** 4

**Summary:**

This paper addresses the challenge of reducing simulation costs in simulation-based inference (SBI). The authors propose a method that leverages the Multilevel Monte Carlo (MLMC) technique for SBI, enabling significant cost reduction by utilizing a sequence of simulations with varying levels of accuracy. Theoretical results support the correctness of the proposed method. The effectiveness of the approach is demonstrated using several simulators, including a real-world cosmology simulator.

**Questions:**

• P115: Regarding the phrase “making NPEs fully amortized” — it would be helpful to clarify what “amortized” means in this context, particularly in comparison to NLE. My understanding is that “amortized” refers to the fact that NPEs do not rely on classical sampling schemes such as MCMC (as used in NLE), thereby avoiding their associated drawbacks.

• I believe Theorem 1 and 2 is a very important result as it shows how the upper bound of the variance depends on the properties of the simulator. My understanding is that the motivation for applying MLMC to SBI lies in its ability to reduce the variance of the loss compared to standard MC. In addition to these theorems, if it can be theoretically shown that the upper bound of MLMC is smaller than that of MC, it would be very interesting.

Regarding the paragraph on the optimisation of the MLMC objective: I understand that this optimisation is a crucial aspect of implementing MLMC. In the current structure, the discussion is placed in Appendix A, but since even the core ideas of the method are important enough, I would hope to see a more detailed discussion included in the main text.

**Ethical Concerns:**

["NO or VERY MINOR ethics concerns only"]

**Final Justification:**

This paper is well-written, and has a good contribution. I vote for acceptance.

**Limitations:**

Yes.

**Paper Formatting Concerns:**

No.

**Quality:**

3

**Strengths And Weaknesses:**

Strength

•  The paper is clearly written and well-structured.

•  It is well-motivated by real-world challenges, addressing a major limitation in SBI: the high computational cost of simulations.

•  In addition to extensive simulation results, the paper includes a detailed theoretical analysis. Some of the theoretical results also provide useful implementation insights, which further support the proposed method.

•  As the authors mention, applying MLMC to modern problems such as SBI or probabilistic numerics is a timely and relevant direction.

Weekness

・The technical and methodological novelty of applying MLMC to SBI may be relatively straightforward.

Taking the weekness into the consideration, the paper’s theoretical contributions—particularly the non-trivial conditions that offer valuable insight into implementation—and solutions to practical challenges, especially in optimization, are substantial enough to outweigh this weekness. Overall, I like this paper, which make me think that this paper deserve accepted in NeurIPS. I decide to assign a score of 5.  Although I am giving an Accept rating prior to the rebuttal phase, I hope that further revisions during the discussion phase will enhance the quality of the paper!

---

> ### Author Rebuttal · Authors · 2025-07-30
>
> Thank you for your positive comments and strong support for our work. We address your comments and questions below.
>
> >The technical and methodological novelty of applying MLMC to SBI may be relatively straightforward.
>
>   We agree that the application of multi-level Monte Carlo to NPE and NLE is straightforward. We see this as an advantage of our approach, since it makes it more likely that the method will be widely adopted. Also, as you pointed out, the extensive theory, experiments, and practical solutions hopefully provide strong contributions towards solving the computational issue of dealing with costly simulators.
>
> ---
>
> >P115: Regarding the phrase “making NPEs fully amortized”—it would be helpful to clarify what “amortized” means in this context, particularly in comparison to NLE.
>
>  We use the phrase "fully amortized" in the context of NPE to mean that obtaining the posterior distribution for a new observed dataset is instantaneous through a simple forward pass of the neural network, once the network is trained. This is to differentiate NPE from NLE, which can be considered to be "partially amortized": although the conditional density $q_\phi(x \mid \theta)$ can be evaluated instantaneously as well for different $\theta$ and $x$ values once trained, obtaining the posterior distribution still requires MCMC schemes. We will make this distinction explicit in the main text.
>
> ---
>
> >I believe Theorem 1 and 2...if it can be theoretically shown that the upper bound of MLMC is smaller than that of MC, it would be very interesting.
>
>  This is a very good point, thank you very much. In fact, our theory already allows for this since the bound for $\text{Var}[h_0(\phi)]$ in Theorem 1 is a bound for an MC-type objective for the high-fidelity simulator if we replace $n_0$ with $n_L$ and $G^0$ with $G^L$ (assuming the MC objective is evaluated only on high-fidelity simulations). By comparing such a bound with the MLMC bound in Equation 3, we can come up with conditions on $n_0, \dots, n_L$ and $G^0, \dots, G^L$ that lead to a smaller MLMC upper bound than that of MC. However, note that this is only a comparison of upper bounds, as opposed to a direct comparison of variances, which is typically out of reach of MLMC theory. We will clarify this interesting point in the camera-ready version.
>
> ---
>
> >Regarding the paragraph on the optimisation...I would hope to see a more detailed discussion included in the main text.
>
>   Agreed. Due to space constraints, we initially had to move some of the discussion to the appendix, but we are planning on moving this back to the main text for the camera-ready version of our paper.
>
> ---
>
> Thank you again for your careful review of our paper. Please let us know if there is anything else we can clarify.

---

> ### Comment · Reviewer_PW7W · 2025-08-02
> **Reply**
>
> Thanks for your reply.
>
> > By comparing such a bound with the MLMC bound in Equation 3, we can come up with conditions on  and  that lead to a smaller MLMC upper bound than that of MC. However, note that this is only a comparison of upper bounds, as opposed to a direct comparison of variances, which is typically out of reach of MLMC theory.
>
> Thanks, this address my concern. I think it is worth noting that MLMC can theoretically yield a tighter bound than standard MC with an appropriate choice of n_i. If possible, it would also be helpful to include some guidance—based on theoretical insights—on how to choose n_i to achieve a smaller bound.
>
> Overall, after reading the rebuttal as well as the other reviews, I have decided to maintain my initial scores in favor of acceptance.

---

> > ### Author Response · Authors · 2025-08-02
> >
> > Yes, we will include a discussion regarding this in the Theory section. Thank you for engaging with us. We really appreciate it.

---

### Official Review · Reviewer_HeBn · 2025-06-23

**Clarity:** 4
**Significance:** 3
**Originality:** 3
**Rating:** 5
**Confidence:** 2

**Summary:**

This paper proposes an approach to more simulation-efficient SBi in the case that multiple simulators with varying cost/fidelity are available, similar to Krouglova et al. The approach is grounded in multilevel Monte Carlo and is applicable to NLE and NPE. In various comparisons, the method is competitive with or outperforms baselines.

**Questions:**

1. Can you decouple the contributions of the two gradient modifications used? How does optimization go if you use only one of them?
2. Given some rough estimates of cost and fidelity, can you provide any heuristics on how to choose the number of simulations for each simulator? Can you show how performance varies as you vary the allocation of a fixed computational budget to e.g., two simulators, one high- and one low-fidelity?

**Ethical Concerns:**

["NO or VERY MINOR ethics concerns only"]

**Final Justification:**

The proposed method is relatively simple, theoretically grounded, addresses an important problem, and it works. My concerns/questions about the gradient modifications and general heuristics for effectively allocating simulation budget were well addressed by the authors' response. I don't have a good sense of how limiting the need for seed matching and structure alignment across fidelities is, as mentioned by reviewer kZeL, but the work is otherwise thorough and clearly presented.

**Limitations:**

Yes

**Quality:**

3

**Strengths And Weaknesses:**

**Strengths:** The proposed method is theoretically grounded and quite straightforward. It performs well in all of the evaluations shown without requiring any additional hyperparameters. The method can be applied to cases where there are more than 2 varying-fidelity simulators and it can also be combined with other existing approaches, so it is quite broadly applicable. The text itself is very clearly written and easy to understand.

**Weaknesses:**
- The question of how to allocate a fixed computational budget to simulations remains unaddressed (though it is arguably beyond the scope of this work). The authors note that computing the optimal samples for each level is not tractable, but some empirical results/guidance on this question would still be useful in my opinion.
- In a few of the experiments (Fig 2a, Fig 5), the proposed method achieves better mean/median scores but also has longer tails with poor performance, suggesting the multilevel objective can introduce higher performance variability across samples than the baselines.
- Though simple, the proposed loss poses problems with optimization that require some gradient hacks.

---

> ### Author Rebuttal · Authors · 2025-07-30
>
> Thank you for your constructive feedback and the time and effort applied in reviewing our manuscript. We address your comments and questions below.
>
> >The question of how to allocate a fixed computational budget to simulations remains unaddressed (though it is arguably beyond the scope of this work). The authors note that computing the optimal samples for each level is not tractable, but some empirical results/guidance on this question would still be useful in my opinion.
>
>  Agreed. Though our theory does not provide explicit results on how to select the $n_0, \dots, n_L$ due to some intractable constants in Theorem 2, it can still provide some useful intuition on how to choose the number of samples at each level.
>
> For instance, consider a two-level simulator. According to our theory, if the low-fidelity simulator is known to be a good approximation of the high-fidelity one, it makes sense to allocate large budget to generating low-fidelity data, while only a small amount of high-fidelity data would be sufficient. On the other hand, if the outputs differ significantly, allocating more budget for high-fidelity data makes more sense, despite the higher cost, to accurately capture the differences. This principle is formalised by our theory and can be extended to simulators with arbitrary fidelity levels. In the toggle switch simulator, we allocated a large budget to approximating the difference between the low- and medium-fidelity simulators, based on prior knowledge that this difference is substantial and worth spending large budget. This lead to improved performance, as we demonstrate with an additional experiment below.
>
> ---
>
> >Can you show how performance varies as you vary the allocation of a fixed computational budget to e.g., two simulators, one high- and one low-fidelity?
>
> We conducted an experiment on the toggle switch simulator with different allocation of $n_0, n_1, n_2$, keeping the total computational cost roughly the same. Apart from the choice of $n_0 = 10,000, n_1 = 500, n_2 = 100$ that we used in the paper (option A) and is guided by our theory, we also include two other alternatives: (B) $n_0 = 9260, n_1 = 200, n_2 = 300$ and (C) $n_0, n_1, n_2 = 1077$. Option B places more budget on learning the difference between the medium and high-fidelity simulator (opposite of option A), and option C allocates equal number of samples for all the terms. The results below indicate that option A is the best performing, indicating performance can be improved by careful assignment of computational budget led by insights from our theory.
>
>
> |                |    Option A           |    Option B         |    Option C         |    only high       |    only medium     |    only low        |
> |:--------------:|:--------------------:|:-------------------:|:-------------------:|:------------------:|:------------------:|:------------------:|
> | **MMD ($\downarrow$)** | $\mathbf{0.16}$  $(0.23)$ |  $0.33$  $(0.26)$   |  $0.29$  $(0.28)$   |  $0.43$  $(0.29)$  |  $0.37$  $(0.45)$  |  $0.59$  $(0.65)$  |
>
> Table 1: Mean and standard deviation of MMD. The results for Option A, only high, medium, and low remain the same as in the paper.
>
> ---
>
> >Can you decouple the contributions of the two gradient modifications used? How does optimization go if you use only one of them?
>
> Good point. As discussed with reviewer kZeL, we have conducted an ablation study on the gradient adjustment. We trained both ML-NPE and ML-NLE using (i) our gradient adjustment approach which involves both rescaling and projection, (ii) only rescaling, (iii) only projection, and (iv) no gradient adjustment (standard training). We then compared the performance of NPE and NLE on the g-and-k and NLE on the toggle switch experiment. Other than the choice of the gradient adjustment, all the training method / hyperparameters, and evaluation methods remain the same. The results are shown in the table below.
>
> |                                |        (i) both         |    (ii) only rescaling    |    (iii) only projection    |       (iv) standard        |
> |:------------------------------:|:-----------------------:|:-------------------------:|:---------------------------:|:--------------------------:|
> | **g-and-k NPE (NLPD ↓)**       |  $\mathbf{-0.30}$  $(0.31)$  |    $-0.21$  $(0.25)$       |     $-0.11$  $(0.11)$        |     $-0.13$  $(0.27)$       |
> | **g-and-k NLE (KLD ↓)**        |     $0.22$  $(0.47)$       |  $\mathbf{0.21}$  $(0.45)$  |      $0.24$  $(0.46)$        |      $0.24$  $(0.28)$       |
> | **Toggle-switch NLE (MMD ↓)**  |  $\mathbf{0.26}$  $(0.25)$  |     $0.50$  $(0.37)$        |      $0.35$  $(0.27)$        |      $0.59$  $(0.31)$       |
>
> Table 2: Mean and standard deviation of the metrics. For g-and-k NPE, $n_0 = 1000$, $n_1 = 100$, $m = 1000$, for g-and-k NLE, $n_0 = 10^4$, $n_1 = 300$, $m = 1$ as in Section 5.1, for toggle-switch NLE, $n_0 = 10000$, $n_1 = 500$, $n_2 = 100$ as in Section 5.3.
>
> We observe that when the MLMC loss diverges under standard training, as is the case for the toggle switch experiment and g-and-k with NPE, our gradient adjustment approach that combines both gradient rescaling and projection yields the best results. In the case of NLE on the g-and-k simulator, the loss does not diverge and standard training is sufficient. In such cases, the gradient adjustment yields similar results as standard training, albeit with an increase in the variance of the metric. We will include this in the limitations. Therefore, we suggest optimising using our gradient adjustment approach when using ML-NLE or ML-NPE as it avoids the need to first detect whether the loss is diverging or not.
>
> ---
> >In a few of the experiments (Fig 2a, Fig 5), the proposed method achieves better mean/median scores but also has longer tails with poor performance, suggesting the multilevel objective can introduce higher performance variability across samples than the baselines.
>
>  Yes, this is true from what we observe. For Figure 2a, higher variance is not necessarily associated with poor performance. Upper tail of the box plot roughly remains the same for all $n, n_1$. This means that as we increase the simulation budget, performance improves consistently—not just on average, but also without any drop in the worst-case performance.
>
> In the case of Figure 5, the longer tail is due to a few outliers. In fact, our method has narrower $50$\%, $90$\%, $95$\%, empirical interval without removing outliers.
> Regarding the variance of ML-NPE, actually it is lower than that of NPE (ML-NPE: $0.78$, NPE: $0.98$). It probably seems like the variance is higher due to the long tail of NLPD for ML-NPE, which is due to a few outliers ($<1$\% of the test dataset). Removing those outliers reduces variance of ML-NPE to $0.60$, however, we reported the results with the outliers for full transparency.
>
> |         |  2.5%       |  5%          |  25%         |  50%            |  75%            |  95%            |  97.5%          |
> |:-------:|:-----------:|:------------:|:------------:|:---------------:|:---------------:|:---------------:|:---------------:|
> | ML-NPE  |  $-2.07$    |  $-1.87$     |  $-1.41$     |  $\mathbf{-0.97}$ |  $\mathbf{-0.24}$ |  $\mathbf{0.58}$  |  $1.08$         |
> | NPE     |  $\mathbf{-2.38}$ |  $-\mathbf{2.20}$ |  $\mathbf{-1.47}$ |  $-0.30$         |  $0.32$          |  $0.73$          |  $\mathbf{0.85}$ |
>
> Table 3: Quantile of NLPD ($\downarrow$)
>
>
> |         |   50%         |   90%         |   95%         |
> |:-------:|:-------------:|:-------------:|:-------------:|
> | ML-NPE  | $\mathbf{3.15}$ | $\mathbf{2.45}$ | $\mathbf{1.17}$ |
> | NPE     | $3.23$          | $2.92$          | $1.80$          |
>
> Table 4: Width of the empirical interval of NLPD ($\downarrow$)
>
> ---
>
> Thank you again for your thorough evaluation of our paper. Please don’t hesitate to let us know if there’s anything else we can clarify.

---

> ### Comment · Reviewer_HeBn · 2025-08-02
>
> Thank you for the response! All of my concerns were addressed by these new experiments and the clarification on the (perceived) performance variability / longer tails. In addition to the clarifications kZeL requested w.r.t. structural assumptions, I think if there is also just a little bit of text added to the main paper regarding the general heuristic you articulate here about allocating budget for simulations, that would be great. That guidance was not apparent to me in reading the original paper, though as someone not as familiar with this kind of work I might have just not grasped the obvious implications of the theory. In any case, I will raise my score.

---

> > ### Author Response · Authors · 2025-08-02
> >
> > Thank you very much for raising your score. We really appreciate it.
> >
> > Indeed, the point about allocation of budget was not clear in the text. We will make sure to add the relevant text right after Theorem 2 about heuristics on how to pick $n_0, \dots, n_L$, and state the structural assumptions in the beginning of Section 3.
> >
> > Thanks again for engaging with us.

---

### Official Review · Reviewer_kZeL · 2025-07-03

**Clarity:** 3
**Significance:** 3
**Originality:** 3
**Rating:** 5
**Confidence:** 4

**Summary:**

This paper extends existing amortized inference methods with a multilevel Monte Carlo (MLMC) estimator. It scopes its contributions through the lens of simulator based inference (SBI), and integrates MLMC both in neural likelihood estimators (NLEs) and neural posterior estimators (NPEs) to improve on sampling budgets for expensive high fidelity simulators. It achieves this by levering cheaper lower fidelity simulators to draw more training samples for the neural artifacts given the same computational budget. Besides introducing the two more efficient gradient estimators it introduces heuristics for allocation simulation efforts across fidelities, and covers a critical set of gradient surgery techniques to stabilize training. It also provides a theoretical convergence analysis of the introduces estimators. The method is empirically validated on a range of SBI benchmarks: g-and-k, Ornstein-Ornstein–Uhlenbeck, a toggle-switch model, and the CAMELs cosmology dataset.

**Questions:**

1. Why was the paper not targeted on an expensive real world simulator experiment? The papers establishing the Toggle-Switch model (sec. 5.3) and Cosmological Simulation (sec 5.4) are from the SBI machine learning literature and samples can be simulated sufficiently quickly even with high fidelity in them. While they can be made more expensive it is less clear whether the more subtle assumptions about alignment between fidelities in the paper translate into a real world setting, and it would have helped to highlight those. The biggest gains would be demonstratable with larger scale simulations from respective sciences, e.g. a full cell simulator or a more expensive astrophysics simulator, in which single samples can take hours or days to be computed. The SBI literature often suffers from picking very simplified simulator experiments that are of little practical interest, yet this paper aims to address one of the biggest practical limitations of SBI based on amortized inference. It would have been sufficient to cover such an experiment and two smaller ones such as g-and-k and maybe the OU-process.

2. Similarly, why is the evaluation in 5.4 for the Cosmological experiment not more convincing? Figure 5 left shows very high variance of ML-NPE and the empirical coverage (right) is very hard to interpret without understanding the specific simulator well. I would expect a more thorough evaluation and clearer gains here.

3. Krouglova et al. 2025 is taken as the main baseline, but as is pointed out in the discussion their approach is somewhat orthogonal. Why was no other multi-level MCMC method or baseline compared?

**Ethical Concerns:**

["NO or VERY MINOR ethics concerns only"]

**Final Justification:**

The paper is an important contribution to simulator based inference. My concerns have been addressed in the rebuttal. The remaining limitations are the need for gradient surgery, and it is unclear how generally it is applicable.

**Limitations:**

yes

**Paper Formatting Concerns:**

-

**Quality:**

3

**Strengths And Weaknesses:**

# Strengths

1. The paper is a straightforward transfer of multi-level estimators to an amortized inference setting as it is common in SBI.

2. The problem is clear and one of the most important limitations of deep leanring based methods, as they are fairly sample inefficient and are not applicable to expensive simulators, fundamentally limiting the applications of state-of-the-art SBI methods.

3. The framework is established with some generality for both NPE and NLE.

4. The experiments demonstrate the benefits for different forms of low-fidelity approximations.


# Weaknesses

Overall the paper would benefit from being more like a recipe for the application of multi-level methods within its ML-NPE and ML-NLE settings.


## 1. Implicit Structural Fidelity Assumptions

MLMC’s success hinges on structural similarity between simulators at adjacent fidelity levels:

* Outputs must be comparable and aligned for the same parameters and randomness.
* Simulators must support shared seeds or correlated randomness.

These assumptions are not clearly articulated in the main text, and their failure modes are not explored. It is not fully clear whether the results only apply to these experiments because they were chosen in this way, or whether they generalize to a large classes of existing simulators.

## 2. Gradient Surgery Requirements

While the MLMC loss is principled, the paper heavily relies on gradient surgery heuristics—especially balancing and orthogonalization of gradients between fidelity levels—to make training stable. These are:

* Presented with no theoretical justification.
* Disconnected from the variance bounds derived in the paper.
* Not evaluated via ablation studies, despite being empirically essential.

This creates a major gap between the theory and practice of the paper.

## 3. Vacuous Variance Bounds

The paper presents formal bounds on loss variance in Section 4, but:

* This does not translate directly to gradient variance, which governs its optimization.
* The optimization dynamics are dominated by gradient imbalance and instability, which are not treated theoretically.

This leaves the theoretical results largely disconnected from the practical concerns addressed through heuristic gradient adjustments. While the contributions of bounds is appreciated their relationship to the rest of the paper should be made clearer.

---

> ### Author Rebuttal · Authors · 2025-07-30
>
> Thank you very much for your effort and time in writing a detailed review.
>
> >Implicit Structural Fidelity Assumptions: Outputs must be comparable and aligned for the same parameters and randomness. These assumptions are not clearly articulated in the main text, and their failure modes are not explored.
>
> We agree that these assumptions should have been more clearly stated, and we will make this discussion more prominent in the method section of the camera-ready version.
>
> MLMC requires coupling between the data from different fidelity levels in terms of having common random numbers and parameters. In cases where this is not possible, MLMC is not applicable, which is a limitation. However, our method is still applicable when there are partial common random numbers between the different fidelities, as is the case with the toggle switch example (dimension of $u$ is $2T + 1$, where $T$ denotes the fidelity parameter). Similarly, our method can also be applied when the low and high-fidelity simulators only partially share the parameters. However, estimating posterior of parameter that is only present in the high-fidelity simulator can be challenging since we only use high-fidelity data in the correction terms of the MLMC loss. We explore this failure mode in the following additional experiment.
>
> We include an additional parameter in the high fidelity simulator, termed "initial value" $\theta_4 = x_0 \sim \text{uniform}([0, 4])$ in Ornstein–Uhlenbeck process (experiment 5.2) with $n_1 = 100$, instead of fixing it at $x_0 = 2.0$. We additionally trained the conditional density estimator using only $n = 100$ data from the high fidelity simulator. All the other settings remain the same.
>
>
> |             |     ML-NPE          |      TF-NPE         |   NPE (high only)    |
> |:-----------:|:-------------------:|:-------------------:|:--------------------:|
> | **NLPD ↓**  |  $\mathbf{-0.18}$  $(0.04)$  |  $-0.89$  $(0.52)$  |  $\mathbf{-1.21}$  $(0.53)$  |
> | **KLD ↓**   |  $\mathbf{0.98}$  $(0.08)$   |  $1.24$  $(0.40)$   |  $1.05$  $(0.14)$            |
> Table 1: Mean and standard deviation of the metrics.
>
> We observe that although our method shows a slight improvement in KLD, it notably underperforms in NLPD compared to using only high-fidelity data. This may be due to our training objective being more unstable and therefore more sensitive to large differences in simulator outputs introduced by the additional parameter $x_0$. We will include this result and discuss the different assumptions in depth in the main text.
>
> ---
> >Gradient Surgery Requirements
>
> We agree that the approach of the optimisation is heuristic without theory, but theory for gradient surgery would be a major project in itself and is therefore beyond the scope of this paper. However, as suggested, we conducted ablation studies on the effect of gradient adjustment and the performance which now provide much more detail on this approach.
>
> We trained both ML-NPE and ML-NLE using (i) our gradient adjustment approach which involves both rescaling and projection, (ii) only rescaling, (iii) only projection, and (iv) no gradient adjustment (standard training). We then compared the performance of NPE and NLE on the g-and-k and NLE on the toggle switch experiment. Other than the choice of the gradient adjustment, all the training method / hyperparameters, and evaluation methods remain the same. The results are shown in the table below.
>
>
> |                                |        (i) both         |    (ii) only rescaling    |    (iii) only projection    |       (iv) standard        |
> |:------------------------------:|:-----------------------:|:-------------------------:|:---------------------------:|:--------------------------:|
> | **g-and-k NPE (NLPD ↓)**       |  $\mathbf{-0.30}$  $(0.31)$  |    $-0.21$  $(0.25)$       |     $-0.11$  $(0.11)$        |     $-0.13$  $(0.27)$       |
> | **g-and-k NLE (KLD ↓)**        |     $0.22$  $(0.47)$       |  $\mathbf{0.21}$  $(0.45)$  |      $0.24$  $(0.46)$        |      $0.24$  $(0.28)$       |
> | **Toggle-switch NLE (MMD ↓)**  |  $\mathbf{0.26}$  $(0.25)$  |     $0.50$  $(0.37)$        |      $0.35$  $(0.27)$        |      $0.59$  $(0.31)$       |
>
> Table 2: Mean and standard deviation of the metrics. For g-and-k NPE, $n_0 = 1000$, $n_1 = 100$, $m = 1000$, for g-and-k NLE, $n_0 = 10^4$, $n_1 = 300$ as in Section 5.1, for toggle-switch NLE, $n_0 = 10000$, $n_1 = 500$, $n_2 = 100$ as in Section 5.3.
>
> We observe that when the MLMC loss diverges under standard training, as is the case for the toggle switch experiment and g-and-k with NPE, our gradient adjustment approach that combines both gradient rescaling and projection  yields the best results.  In the case of NLE on the g-and-k simulator, the loss does not diverge and standard training is sufficient. In such cases, the gradient adjustment yields similar results as standard training, albeit with an increase in the variance of the metric. Therefore, we suggest optimising using our gradient adjustment approach when using ML-NLE or ML-NPE as it avoids the need to first detect whether the loss is diverging or not. We will naturally include these results and the associated discussion in the main text.
>
> We note that the optimisation requires some form of regularization, and the one we used is one possible choice. According to our experiments, gradient adjustment performed the best among the options we tried (e.g., regularizing the contribution of the difference term when it dominates the first term, or penalizing high variance in the difference term). We will clarify this in the main text.
>
>
> ---
>
> >The paper presents formal bounds on loss variance in Section 4, but This does not translate directly to gradient variance, which governs its optimization.
>
> Thank you for the insightful point; we completely agree with you that a bound on the gradient would be interesting, and thankfully this is not very difficult to derive from the existing proof technique. This only requires replacing (A3) so that it holds for the gradient of the log-conditional density ($\nabla_\phi \log q_\phi^\text{NLE}(x | \theta)$ for NLE and $\nabla_\phi \log q_\phi^\text{NPE}(\theta | x_1,\dots x_m)$) rather than the log conditional density itself. The rest of the proof is almost identical. We will now include this additional result in the paper.
>
> ---
>
> >Why was the paper not targeted on an expensive real world simulator experiment?
>
> Thank you for the suggestion. Actually, the CAMELS simulations suite we used in Section 5.4 is one of the most expensive cosmological suites ever run (a small fraction of the next generation are being run on the UK DIRAC HPC Facility with 15M CPUh). These are real state-of-the-art physics simulations being used for SBI analyses with real-world (observational) data; see [4]. This was clearly poorly explained, and we will remedy this for the camera-ready version.
>
> ---
>
> >Similarly, why is the evaluation in 5.4 for the Cosmological experiment not more convincing? Figure 5 left shows very high variance of ML-NPE and the empirical coverage (right) is very hard to interpret without understanding the specific simulator well. I would expect a more thorough evaluation and clearer gains here.
>
>  We agree that our analysis of the results from the cosmological simulations is limited, primarily because of the low amount of data, but we did use a state-of-the-art simulator. We meant our results as illustrations that combining multifidelity simulators with our method leads to better performance, and we refer the readers to recent works [3-4] for a more in-depth illustration of the benefits of multifidelity in this context.
>
> Regarding the variance of ML-NPE, actually it is lower than that of NPE (ML-NPE: $0.78$, NPE: $0.98$). It probably seems like the variance is higher due to the long tail of NLPD for ML-NPE, which is due to a few outliers ($<1$\% of the test dataset). Removing those outliers reduces variance of ML-NPE to $0.60$, however, we reported the results with the outliers for full transparency.
>
> ---
>
> >Krouglova et al. (2025) is taken as the main baseline, but as is pointed out in the discussion their approach is somewhat orthogonal. Why was no other multi-level MCMC method or baseline compared?
>
>    For amortized SBI methods, Krouglova et al. 2025 is the only existing multi-fidelity method, although [5] came out just as we were submitting this paper to NeurIPS (this is also a two-stage approach like Krouglova et al. 2025). Other alternatives include the multi-fidelity ABC methods referenced in the paper. However, such a comparison would not lead to conclusions about the different multi-fidelity approaches as ABC (or any other sampling based SBI method) is known to yield very different posteriors than NPE/NLE for the same amount of data. Moreover, it is not amortised, so computing performance metrics over multiple observed datasets is computationally expensive. If the reviewer has any particular method in mind then we would be happy to add some comparison.
>
> ---
> **References:**
>
> [1] Krouglova, Anastasia N., et al. "Multifidelity simulation-based inference for computationally expensive simulators." arXiv preprint arXiv:2502.08416 (2025).
>
> [2] Yu, Tianhe, et al. "Gradient surgery for multi-task learning." Advances in neural information processing systems 33 (2020): 5824-5836.
>
> [3] Thiele, Leander, Adrian E. Bayer, and Naoya Takeishi. "Simulation-Efficient Cosmological Inference with Multi-Fidelity SBI." arXiv preprint arXiv:2507.00514 (2025).
>
> [4] Lovell, Christopher C., et al. "Learning the Universe: Cosmological and Astrophysical Parameter Inference with Galaxy Luminosity Functions and Colours." arXiv preprint arXiv:2411.13960 (2024).
>
> [5] Tatsuoka, Caroline, et al. "Multi-fidelity parameter estimation using conditional diffusion models." arXiv preprint arXiv:2504.01894 (2025).
>
> ---
>
> Thank you again for your careful assessment. Please let us know if there is anything else we can clarify further.

---

> > ### Comment · Reviewer_kZeL · 2025-08-05
> >
> > Thank you for the comprehensive response! I appreciate this line of work and applications to expensive scientific simulations. My concerns have been addressed and will raise my score.

---

### Comment · Area_Chair_N1AD · 2025-08-03

Dear authors and reviewers,

First of all, thank you all for your efforts so far. The author-reviewer discussion period will end on August 6.

@Authors: If not done already, please answer all questions raised by the reviewers. Remain factual, short and concise in your responses, and make sure to address all points raised.

@Reviewers: Read the authors' responses and further discuss the paper with the authors if necessary. In particular, if the concerns you raised have been addressed, take the opportunity to update your review and score accordingly. If some concerns remain, or if you share concerns raised by other reviewers, please make sure to clearly state them in your review. In this case, consider updating your review accordingly (positively or negatively). You can also maintain your review as is, if you feel that the authors' responses did not address your concerns.

I will reach out to you again during the reviewer-AC discussion period (August 7 to August 13) to finalize the reviews and scores.

The AC

---

> ### Comment · Area_Chair_N1AD · 2025-08-08
>
> Dear reviewers,
>
> The reviewers-authors discussion phase will end in less than 24 hours.
>
> If not done already, make sure to submit the "Mandatory Acknowledgement" that confirms that you have read the reviews, participated in the discussion, and provided final feedback in the "Final justification" text box.
>
> Be mindful of the time and efforts the authors have invested in answering your questions and at least acknowledge their responses. Make sure to provide a fair and scientifically grounded review and score. If you have changed your mind about the paper, please update your review and score accordingly. If you have not changed your mind, please provide a clear and sound justification for your final review and score.
>
> Best regards,
> The AC

---

### Note · Authors · 2025-08-13

We would like to thank all the reviewers and the AC for their engagement in this review process. The initial feedback was overwhelmingly positive (scores: 5, 4, 4), and following the rebuttal and discussion stages, all the reviewer concerns were fully addressed, leading several reviewers to raise their scores. All reviewers now give a very strong backing to this paper ('5 – Accept').

---

### Decision · Program_Chairs · 2025-09-17

**Decision:**

Accept (poster)

**Comment:**

The average rating is 5, with all reviewers unanimously recommending acceptance (5, 5, 5). We encourage the authors to implement the modifications discussed with the reviewers in the final version of the paper.

Recommendation: acceptance.